# The microscopic mechanism of water immersion and collapsibility in Malan loess with different particle size

Li Li[1,2☯], Qinglong Zhang[1,2☯¤a], Huandong Mu🅾[3‡¤b*], Longhao Zheng[3], Yisong Bai[3]

**1** Gansu Provincial Transportation Research Institute Group Co., Ltd., Lanzhou, Gansu, China, **2** Gansu Provincial Center of Technology Innovation for Health Monitoring and Security Assessment in Bridge and Tunnel, Lanzhou, Gansu, China, **3** School of Civil Engineering and Architecture, Xi`an University of Technology, Xi`an, Shaanxi, China

☯ These authors contributed equally to this work.
‡ HM also contributed equally to this work.
¤a Current Address: No. 213, Jiuquan Road, Chengguan District, Lanzhou, Gansu, China
¤b Current Address: No. 5, Jinhua South Road, Xi'an, Shaanxi, China
* mhdyhx@xaut.edu.cn

## Abstract

Collapsibility of loess is a widespread, highly destructive geological hazard on the Chinese Loess Plateau. Malan loess exhibits distinct regional particle size variations, but the collapsible deformation characteristics and underlying microscopic mechanisms of loess with different particle sizes remain insufficiently understood. This study selected sandy (Jingbian), silty (Yan'an), and clayey (Jingyang) Malan loess in Shaanxi as representative samples to investigate collapsible deformation and clarify intrinsic mechanisms. Results show particle size and clay content significantly affect loess' physical-mechanical properties: particle shape transitions from angular to sub-rounded/rounded, with clay distributing as adhesion (sandy), bridging (silty), or filling (clayey). Collapse is dominated by clay softening, skeleton destruction, and void filling. Post-collapse, macropores (>50 μm) convert to mesopores (2–50 μm), porosity drops ~10%, and pore orientation homogenizes. Generalized collapse mechanism models for different particle size Malan loess are proposed, providing a theoretical basis for hazard mitigation.

## Introduction

Loess is a type of soil sediment composed of soil skeleton particles with varying sizes, characterized by large pores and weak cohesion. Both the Late Pleistocene $Q_3$ loess (Malan loess) and Holocene $Q_4$ loess (loess-like soil) exhibit collapsibility [1], The collapsibility of loess often leads to severe geohazards (Fig 1), such as the formation of sinkhole under the infiltration of surface water and the destruction of pavement. Subsequent slope collapse threatening the safety of production and life; Uneven settlement will cause damage to roads and building foundations. Loess is

**Data availability statement:** All relevant data are within the manuscript and its Supporting Information files.

**Funding:** This work was supported by the National Natural Science Foundation of China (42372336 to HM), the Special Fund for Basic Scientific Research of Central Universities (300102262505 to HM) and the 2025 State-owned Capital Operating Budget to Support Scientific Research Projects of Provincial Enterprises (2025GZ017 to QZ).

**Competing interests:** The authors have declared that no competing interests exist.

widely distributed around the world. As the largest distribution area of Malan Loess in China, the Loess Plateau shows a clear zonal pattern in loess particle composition, with sandy loess, silty loess, and clayey loess distributed sequentially from northwest to southeast [2,3]. These different types of loess, owing to variations in the geographical, geological, and climatic conditions of their respective regions, exhibit differ significantly in sediment thickness, stratigraphic features, physical and mechanical properties, and deformation characteristics [4–9]. The morphology of loess aggregates varies with particle size, which in turn affects the soil's microstructure and collapsibility. Investigating the collapsibility and deformation of Malan loess with different particle sizes is critical for elucidating its underlying mechanism.

Scholars all over the world have conducted continuous research on the collapsibility of loess. Since the 1930s, numerous studies have focused on the loess collapse hypothesis, collapse coefficient and collapse initial pressure, collapse deformation, and collapse susceptibility evaluation [10–14]. As research delves further, scholars both at home and abroad have conducted extensive studies on the collapse mechanism of loess, proposing theories such as the capillary theory [15], dissolution of salts theory [16], insufficient colloids theory [17], wedge entry of water film theory [18], and structural theory [19–25]. Among them, Among these, the structural theory integrates the inherent structural characteristics of loess with external influencing factors, providing a systematic mechanical explanation for the collapsible deformation mechanism of loess.. Terzaghi [26] was the first to investigate the microstructure of soil, Currently, significant progress has been made in explaining loess collapsibility mechanisms through microstructure analysis [27–30], Scholars generally concur that the fundamental cause of loess collapsibility is the presence of open-framework pores, with pore size distribution being closely correlated with collapsibility [31]. Sand loess, silty loess and clay loess are three typical Malan loess, with substantial differences in silt and clay contents.. Their content and sedimentary form directly affect the macroscopic collapsibility and deformation characteristics of loess.

Previous studies have systematically explored the mechanism of loess collapsibility and its microstructure, yielding notable findings. However, loess microstructure primarily depends on particle bonding and cementation, which vary with the sand and clay contents of the soil. For example, the loess with different sand and clay content can form a variety of structural types, such as skeleton structure, flocculation structure, agglomerate structure, granular structure, coagulated structure, lamellar structure, honeycomb structure, spongy structure, magnetic domain structure, matrix structure, etc. Among them, the skeleton structure is mainly composed of silt particles, and the clay particles are distributed unevenly between the skeleton, and most of them form large pores. The flock-like structure is mainly composed of clay-dominated flocs as the basic structural units, and the clay minerals are mostly arranged in edge-plane, edge-edge and a small amount of face-plane, so the engineering geological properties are relatively uniform. Most of the clay mineral sheets with honeycomb structure exist in a face-side arrangement, forming a chain with dense pores, high compressibility, low strength and no anisotropy. Despite the significant differences in silt, sand, and clay contents of these three typical Malan loess types—and the consequent variations in

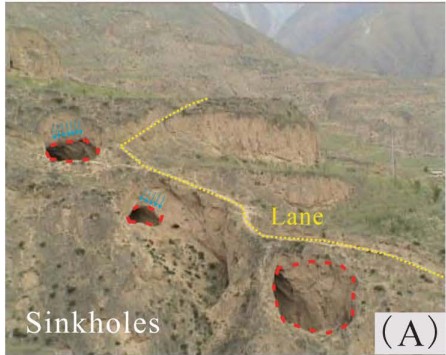 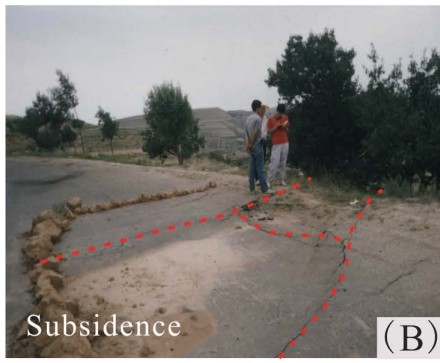 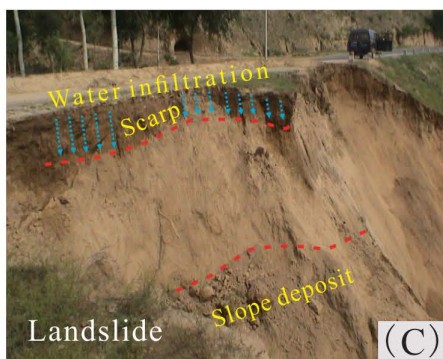

**Fig 1. Damage of loess collapsibility. (A)** Cascade sinkhole caused by loess collapsibility; **(B)** Loess highway collapse; **(C)** Loess slope landslide.

structural types that affect macroscopic collapsibility and deformation—current studies have overlooked the differences in engineering geological properties caused by structural type variations. Additionally, comparative studies on the microstructure and collapsibility of loess with different silt and clay contents remain scarce.

Therefore, this study selected three typical Malan loess with different particle size components as the research object. Through basic physical and mechanical tests, particle size analysis, XRD tests, collapsibility tests and scanning electron microscopy tests, the material composition, physical and mechanical properties, collapsibility characteristics, morphological characteristics and microstructure types of Malan loess with different particle sizes were compared and analyzed. Based on SEM observations, we examined the microstructure of fine-grained loess, directly observing mineral composition, crystallization status, interparticle contact relationships, and structural unit morphologies to determine the structural type of each fine-grained loess sample. On this basis, the collapsibility mechanism of loess with different microstructure types is revealed. Finally, integrating its morphological characteristics and microstructure types, a generalized collapsibility mechanism model of Malan loess with different grain sizes was proposed to provide a theoretical basis for the collapsibility evaluation of Malan loess with different grain sizes.

## Materials and methods

### Materials

No specific permits were required for the collection of loess samples in this study. All sampling locations are in open natural areas, which are not part of protected areas, private lands, or other restricted areas. The sampling process only involved small-volume surface soil collection, which did not cause damage to the local ecological environment, alter the original geological profile, or violate any relevant national or local administrative regulations (the "Regulations of the People's Republic of China on Nature Reserves" and local ecological protection policies).

The loess on the Chinese Loess Plateau exhibits distinct zonal distribution characteristics, which are classified into sandy loess, silty loess, and clayey loess in sequence from the northwest to the southeast. The loess samples analyzed in this study were collected from three Late Pleistocene (Q3) Malan loess sedimentary areas with unique geological features, as detailed below:

Jingbian County (mid-temperate semi-arid continental monsoon climate, annual precipitation of 380–420 mm): The sampling site is located in the aeolian loess hilly region in the northwestern part of the Chinese Loess Plateau. Due to the significant influence of the northwest winter monsoon during the sedimentation process, the loess sediments in this region are dominated by sandy components, making it a typical profile of sandy Malan loess in arid areas.

Yan'an City (warm-temperate semi-humid to semi-arid transitional climate, annual precipitation of 500–550 mm): The sampling site is situated in the loess ridge and gully region in the central part of the Chinese Loess Plateau. The particle composition is mainly silt, rendering it a representative profile of silty Malan loess in the transitional climate zone.

Jingyang County (warm-temperate semi-humid monsoon climate, annual precipitation of 580–620 mm): The sampling site is located in the loess tableland region in the southern part of the Chinese Loess Plateau. The sedimentary environment of Malan loess here is relatively stable, and the clay content is significantly higher than that in the northern regions (attributed to the moist air currents brought by the summer monsoon), making it a typical outcrop of clayey Malan loess in semi-humid areas.

To ensure regional representativeness of samples and reduce internal variability within a single sampling site, this study set 3 parallel sub-sampling points (arranged in a plum-blossom pattern with a spacing of 5–8 meters) at each of the three main sampling locations (Jingbian, Yan'an, and Jingyang), resulting in a total of 9 undisturbed soil samples (3 loess types × 3 parallel sub-samples per type). All samples were collected from the same Late Pleistocene (Q3) Malan loess stratum at a depth of 3–4 meters to avoid the impact of stratigraphic differences. This multi-sub-sampling design can effectively eliminate the interference of local microtopographic variations on sample properties, ensuring that the tested loess characteristics are representative of the typical attributes of the corresponding loess type (see Fig. 2 for specific sampling points). Samples were collected manually through a process of first excavating undisturbed soil blocks by artificial slotting, followed by cutting and encapsulating the soil samples with PVC pipes (110 mm in diameter) — this PVC pipe encapsulation method is critical for maintaining sample integrity during storage and transportation. After encapsulation, paraffin-coated cardboard was used to cover both ends of the PVC pipes, which were then sealed with adhesive tape; meanwhile, the sampling location, upper/lower stratigraphic positions of the samples, and sampling date were clearly marked on the pipe surface. To further ensure sealing performance, all gaps (including the joints between PVC pipes and paraffin-coated cardboard) were subjected to secondary wax sealing. Prior to laboratory tests, the PVC-encapsulated

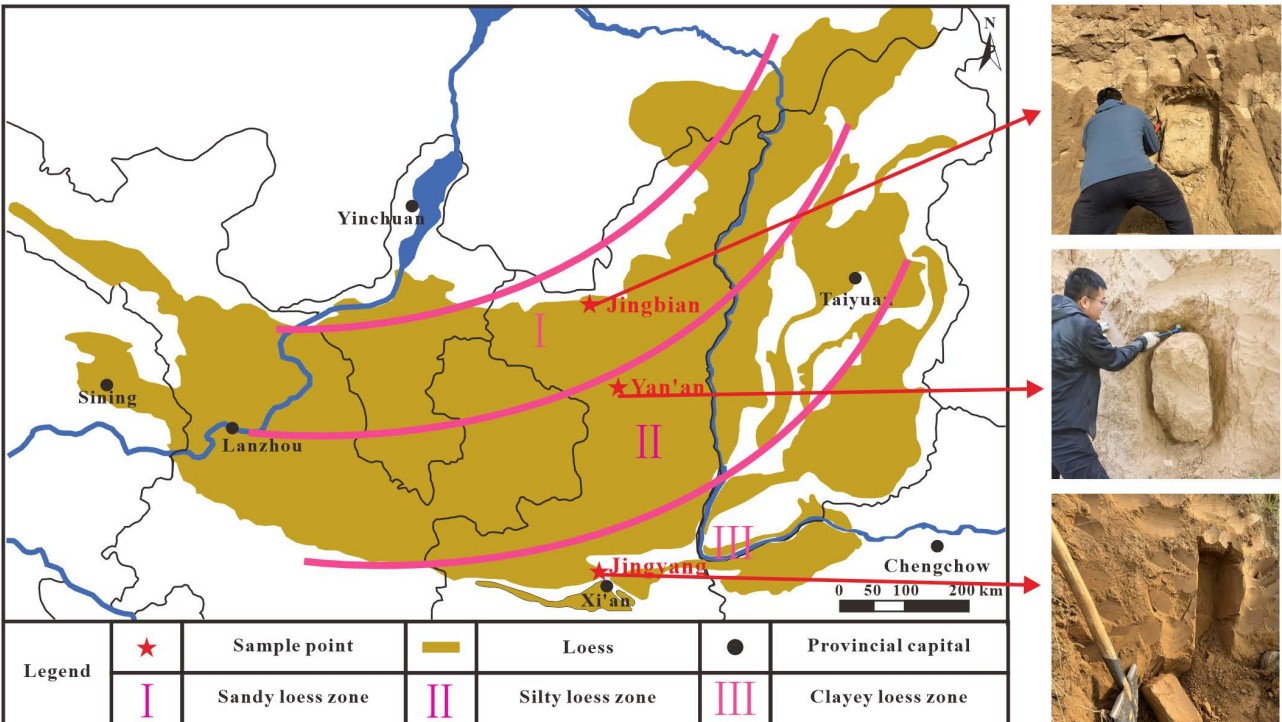

**Fig 2. Map of sampling locations for Malan loess on the loess plateau.**

samples were stored in a controlled environment with a temperature of 15–25°C (to prevent thermal expansion and contraction of the soil matrix) and a relative humidity (RH) of 40–60% (to avoid moisture absorption or desiccation of the undisturbed loess). During transportation, the PVC-encapsulated undisturbed soil samples were protected from vibration, freezing (temperature <0°C), or excessive wetting (RH > 70%), handled with care, and prevented from being inverted to avoid disrupting the original stratigraphic order of the samples.To clarify the physicomechanical properties of the samples, laboratory tests were conducted on undisturbed soil samples, with the core methods as follows: Natural density was determined by the ring knife method; both dry density and natural moisture content were measured by the 105~110°C oven-drying method, calculating the ratios of dry soil mass to ring knife volume, and water mass to dry soil mass, respectively; the relative density of soil particles was determined by the pycnometer method.

Liquid limit was measured by the disc liquid limit tester method, i.e., the moisture content when a 13 mm crack appears in the soil sample; plastic limit was determined by the thread rolling method, i.e., the moisture content when a 3 mm soil thread just breaks. Particle size analysis was performed by a combination of sieving method (for coarse particles) and hydrometer method (for fine particles). After obtaining the particle size distribution curve, the uniformity coefficient ($C_u = d_{60}/d_{30}$), curvature coefficient ($C_c = d^2_{30}/(d_{10} \times d_{60})$), and clay content (particle size < 0.005 mm) were calculated.

Compressibility indices were obtained by consolidation tests to generate e-p curves. The compression coefficient was calculated using the formula ($a = (e_1-e_2)/(p_2-p_1)$). Subsequently, the compression modulus was computed by the formula ($E_s = (1+e_0)/a$) in combination with the initial void ratio $e_0$. The physicomechanical parameters of Malan loess with different particle sizes are summarized in Table 1.

## Sample preparations and testing methods

For the particle size analysis, a Bettersize 2000 laser particle size analyzer was employed. X-ray diffraction (XRD) analysis was conducted using a Puxiang XD-3 automatic X-ray powder diffractometer. The consolidation and collapsibility tests were carried out using a WG-type single-lever consolidometer. The scanning electron microscopy (SEM) test was conducted with a JSM-7610F field emission scanning electron microscope, and the quantitative analysis of SEM images was performed using the PCAS system. The detailed experimental setup and instrumentation configuration are presented in Fig 3.

## Grain distribution analysis test

For loess sample preparation, representative undisturbed loess samples were collected and dispersed on a rubber plate using a wooden roller. The dispersed loess is then passed through a 2-mm sieve. Subsequently, deionized water and

**Table 1. Basic physico-mechanical indices of Malan loess with different particle sizes.**

| Indicators/soil samples | Sandy loess | Silty loess | Clay loess |
|---|---|---|---|
| Natural density (g/cm³) | 1.72 | 1.65 | 1.53 |
| Dry density (g/cm³) | 1.59 | 1.52 | 1.33 |
| Natural moisture content (%) | 7.83 | 8.33 | 14.86 |
| Relative density of soil particles | 2.69 | 2.70 | 2.75 |
| Liquid limit (%) | 25.3 | 27.7 | 31.5 |
| Plastic limit (%) | 15.8 | 17.3 | 18.8 |
| Uniformity coefficient | $C_u = 5.5$ | $C_u = 6.7$ | $C_u = 7.7$ |
| curvature coefficient | $C_c = 1.48$ | $C_c = 1.45$ | $C_c = 1.55$ |
| Compression coefficient | 0.59 | 0.55 | 0.54 |
| Compression modulus | 3.10 | 3.23 | 3.22 |
| Clay content | 8.67% | 16.29% | 27.87% |

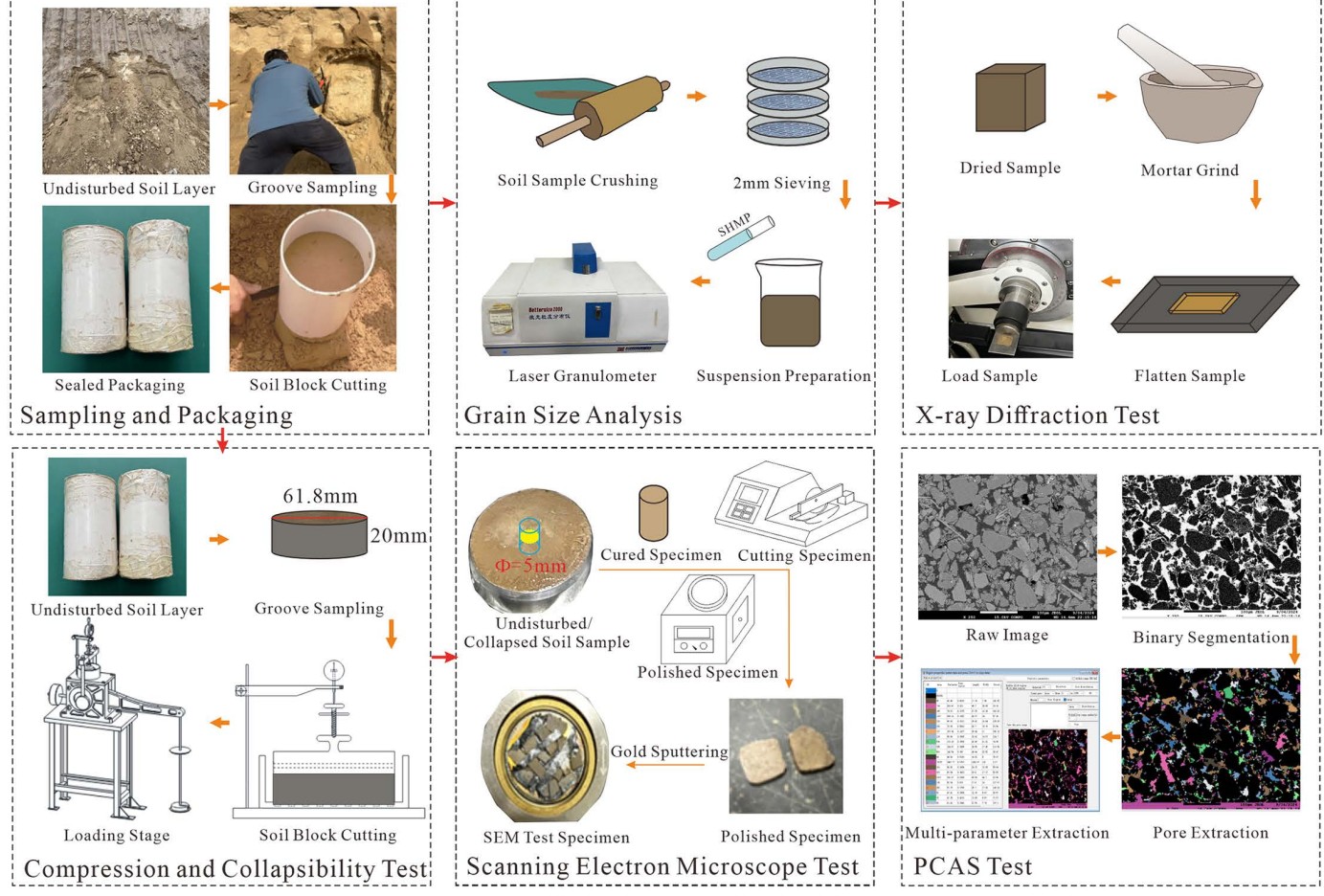

**Fig 3. Schematic workflow of sample preparation and testing for Malan loess with different particle sizes.**

sodium hexametaphosphate (0.5–1 g per liter of deionized water, used as a dispersant) were added, and the mixture was stirred at 300 rpm for 5 minutes to form a homogeneous suspension.

After completing sample preparation, start the Bettersize2000 laser particle size analyzer. Create a new project and set parameters. After water intake and defoaming, perform a background test. When the "light shading rate" prompt appears, add the suspension to the circulation pool in multiple small portions, controlling the light shading rate at 10%–15%. After dispersion stability is achieved, click "real-time" for observation, then perform "continuous testing" to save results and print the report. For automatic testing, set the Standard Operating Procedure parameters and add samples as prompted; the system will automatically complete the subsequent processes.

### X – ray diffraction test

The specific procedures for the X-ray diffraction (XRD) test are as follows: First, weigh approximately 0.5 g of the dry sample (dried at 105±2 °C for 24 h to remove free water) into an agate mortar and grind it until no granular sensation is felt. The ground powder was placed into the groove of a glass sample holder (depth: 1 mm, inner diameter: 20 mm)., compact it with a glass slide, and rotate it 90 ° each time to avoid particle orientation, ensuring a flat sample surface and labeling it.

Subsequently, turn on a series of equipment power supplies, including the main wall power supply and circulating water power supply. Enter the application program, activate the high-voltage power supply of the X-ray generator, open the chamber door to place the sample stage, and then close the chamber door. Initialize the drive in the program interface, set parameters such as detection angle and scanning speed, and start collecting spectral lines. After data collection is complete, save the spectral data to the hard disk first and then export it as needed.

**Loess consolidation test and collapse test**

The undisturbed loess sample test includes: one-dimensional compression test and collapsibility test.

Both tests employed undisturbed sampling using a stainless-steel ring cutter. First, Vaseline was evenly applied to the inner wall of the ring cutter to reduce friction between the soil and the cutter, The ring cutter was then pressed vertically into the undisturbed loess to obtain a soil column, after which the excess soil at both ends of the cutter was trimmed with a wire saw. For the collapsibility test, the ring cutter used has specifications of 61.8 mm in diameter and 20 mm in height, and the initial height (H) of the soil sample is recorded. The shared installation procedure involves placing the sample into the consolidometer, sequentially adding permeable stones and filter paper, and installing a dial gauge to monitor sample deformation.

The two tests shared the same sample installation procedure for the consolidometer (WG-type single-lever oedometer): the ring cutter with the soil sample was placed into the consolidometer cell first; then, a layer of filter paper and a permeable stone were sequentially placed on both the top and bottom of the ring cutter to ensure water permeability. Finally, a dial gauge (precision: 0.001 mm) was mounted above the sample to monitor vertical deformation during loading.

Loading Stage: Compression Test: A preload of 25 kPa is first applied, followed by gradual loading at 50, 100, 200, 400, and 800 kPa. Each load level is maintained until deformation stabilizes (24 hours or a deformation rate < 0.01 mm/h), with the compression amount recorded. Collapsibility Test: Loading is conducted in stages from 25 to 1600 kPa. At each stage, the sample is stabilized until the deformation rate ≤ 0.01 mm/h, and the height (H) is recorded. After the last load stage stabilizes, the sample is saturated by immersion and observed for at least 48 hours until collapsible deformation stabilizes, with the final height ($H_1$) recorded.

Data Calculation and Analysis: For the compression test, the void ratio is calculated based on the compression amount, and a compression curve is plotted to derive the compression coefficient ($a_v$) and compression modulus ($E_s$) for evaluating the compression characteristics.

For the collapsibility test, the collapsibility coefficient ($\delta_s$) is calculated using the following formula:

$$\delta_s = \left(H_0 - H_1\right)/H_0 \tag{1}$$

In the formula: $\delta_s$—collapsibility coefficient, $H_0$—Original height, $H_1$—final height.

This calculation method conforms to the definition of the collapsibility coefficient specified in Code for Building Construction in Collapsible Loess Regions (GB 50025−2018), a national technical standard of China. The standard defines δs as the core index for evaluating loess collapsibility, which provides a direct basis for classifying collapsibility grades and guiding geotechnical engineering design in loess regions.

**SEM Test**

The following are the basic procedures for scanning electron microscopy (SEM):

(1) SEM Sample Preparation: To ensure that the sample can truly reflect the soil structure, the specimen should first be preserved by freezing, followed by air-drying at a constant temperature of 20°C. After air-drying, a cylindrical soil unit with a diameter of 5 mm is extracted from the specimen, and then cured using a mixed curing agent. The components of the mixed curing agent are prepared in a mass ratio of epoxy resin: acetone: ethylenediamine: dibutyl

phthalate = 100:200:7:2. After preparation, the curing agent must be fully stirred until it forms a homogeneous, transparent, and viscous liquid with no obvious delamination, particle agglomeration, local sedimentation, and uniform color; only then can it be used for soil sample curing. Subsequently, the soil sample mixed with the curing agent was placed in the same environment as that for air-drying for curing.

(2) Post-Curing Processing: After curing, the specimen was subjected to sequential cutting (adopting the sectional grinding method), grinding with 500-grit sandpaper, and polishing (reverse polishing for 6–8 minutes using a metallographic polisher) to produce a cylindrical thin section with dimensions of 5 mm (diameter) × 1 mm (thickness).

(3) Gold Coating and Imaging: A gold film is sputtered onto the surface of the prepared cylindrical thin-section specimen to enhance its conductivity. The SEM test should be conducted promptly (within 30 minutes) after gold coating to prevent oxidation of the gold film and surface contamination from air exposure.

## PCAS Test

The Particle and Crack Image Recognition and Analysis System (PCAS) is a specialized automated image processing software designed for intelligent recognition and quantitative analysis of high-resolution particle, pore, and crack images acquired via electron microscopy.. Following binarization processing, the system automatically eliminates noise points, segments and identifies target objects (particles, pores, cracks), and outputs key geometric parameters (area, perimeter, length, shape factor, curvature coefficient) and statistical parameters (porosity, fractal dimension, probability entropy, pore size distribution index). Compared with manual measurement, PCAS offers distinct advantages of high automation, high measurement accuracy, and excellent reproducibility, making it widely applicable to quantitative microstructure research in geotechnical engineering, materials science, and related disciplines.

For the quantitative pore analysis of loess SEM micrographs, strict sample selection criteria must be implemented prior to imaging to ensure the reliability and representativeness of the analytical results: ① Sampling and sample preparation should be prioritized in homogeneous soil regions to avoid the interference of local structural anomalies on the statistical results of pore parameters; ② At least 3 parallel samples should be prepared for each test group, and non-repetitive visual fields should be randomly selected from each sample for imaging analysis to reduce accidental errors caused by a single sample or visual field. The micro-pore structural characteristics of the samples selected in accordance with the above specifications can reflect the structural properties of the overall soil to a certain extent.

For the quantitative analysis of the pores in loess SEM micrographs, the operation procedure is as follows: First, import the SEM micrograph via the "File> Open" function. The system automatically pop up the "Segmentation" segmentation setting window; adjust the Threshold slider to optimize the contrast between pores and the soil matrix, then enable the "Reverse" option to ensure pores are labeled as white and the soil matrix as black. After returning to the main interface, set the pore recognition parameters (Element radius = 2.1, Minimum area = 50) and click "Auto Analysis" to initiate automatic analysis. Within a few seconds, the software generates a "Region Properties" data table containing key geometric parameters (pore area, shape factor), enabling rapid quantitative characterization of the pore structure.

## Fractal dimension

The fractal dimension is an key geometric parameter for characterizing the complexity of t porous media microstructures, and its value exhibits a positive correlation with the irregularity of particle-pore interface geometries. Based on t fractal geometry theory, the fractal dimension of the pore boundary can be quantitatively characterized via the perimeter-area scaling relationship, which adheres to the following formula:

$$LnP = (D/2) \times LnA + C \tag{2}$$

In the formula: D—fractal dimension, P—perimeter, A—area, and C—constant.

This formula indicates that in a log-log coordinate system, there is a linear relationship between the perimeter and the area, and the slope D/2 directly reflects the self-similarity characteristics of the microstructure. After extracting pore geometric parameters (area and perimeter) via the PCAS system, the fractal dimension D can be accurately calculated by performing linear regression analysis on the data using the least squares method. The fractal dimension typically ranges from 1 (for a simple, smooth boundary) to 2 (for a highly complex, space-filling structure). Its quantitative results provide a basis for linking microstructural evolution to macroscopic properties, such as loess collapsibility mechanisms.

## Results and discussion

This study systematically investigated the particle size distribution, mineral composition, collapsible deformation, and microstructural characteristics of three typical Malan loess types (sandy loess, silty loess, and clayey loess). The findings, coupled with in-depth discussions on their underlying mechanisms and engineering implications, are presented as follows:

### Particle size distribution characteristics

The particle size distribution (PSD) and gradation curves of the three Malan loess samples are illustrated in Fig 4. The particle size range of silt and clay fractions in all samples is 0.1–100 μm, covering silt and clay particles; the full particle size range of the samples is 0.1–200 μm, including a small amount of fine sand particles.. The slope of the cumulative distribution curve reflects the concentration of particles within a specific size range, with steeper slopes indicating higher proportions and more pronounced variations in particle distribution. Notably, distinct disparities exist among the three loess types: Sandy loess is dominated by coarse-grained particles (fine sand: 100–200 μm and coarse silt: 50–100 μm), with a frequency curve peaking at 50–100 μm and a sharply rising cumulative curve in the coarse-grained segment, an attribute attributed to its aeolian genesis under arid climatic conditions where strong winds transported coarser particles and limited chemical weathering preserved their coarse-grained nature, resulting in a high coarse particle content, simple composition, and loose skeleton-supported structure. Silty loess displays transitional features, characterized by a broad unimodal frequency curve (peak at 20–50 μm) and a uniformly sloped cumulative curve, indicating a mixed composition dominated by silt with moderate clay and fine sand contents and relatively favorable gradation—an outcome of moderate wind transport distances and balanced deposition of particles with varying sizes. Clayey loess is predominated by fine-grained particles (clay: < 5 μm and fine silt: 5–20 μm), with a frequency curve peak at 20–30 μm and a steeply rising cumulative curve in the fine-grained segment, where the high fine particle content and dense structure are associated with longer aeolian transport distances and deposition under more humid climatic conditions that facilitated particle weathering and refinement.

### Mineral phase analysis of malan loess with different particle sizes

The PSD and gradation curves of loess samples with varying clay contents are presented in Fig 5, demonstrating that fine particles dominate all samples, with particle fineness increasing as clay content rises from the northwest to the southeast of the study area. X-ray diffraction (XRD) analysis reveals that all three loess types contain common minerals (quartz, albite, muscovite, and calcite) but differ significantly in mineral content: Jingbian sandy loess exhibits an extremely high quartz peak intensity and low contents of other minerals, Yan'an silty loess shows higher peak intensities of certain minerals compared to sandy loess, and Jingyang clayey loess displays more balanced mineral peak intensities and a relatively elevated clay mineral content. These mineral composition characteristics are closely linked to particle size: Quartz, a hard and chemically stable mineral, is concentrated in coarse-grained sandy loess to form the soil's skeleton, while clay minerals (illite, smectite) are more abundant in fine-grained clayey loess to enhance interparticle cementation. The gradual increase in clay mineral content with decreasing particle size directly modifies the soil's physical and chemical properties (plasticity, water absorption, and cementation), laying the foundation for differences in collapsibility. The consistency in

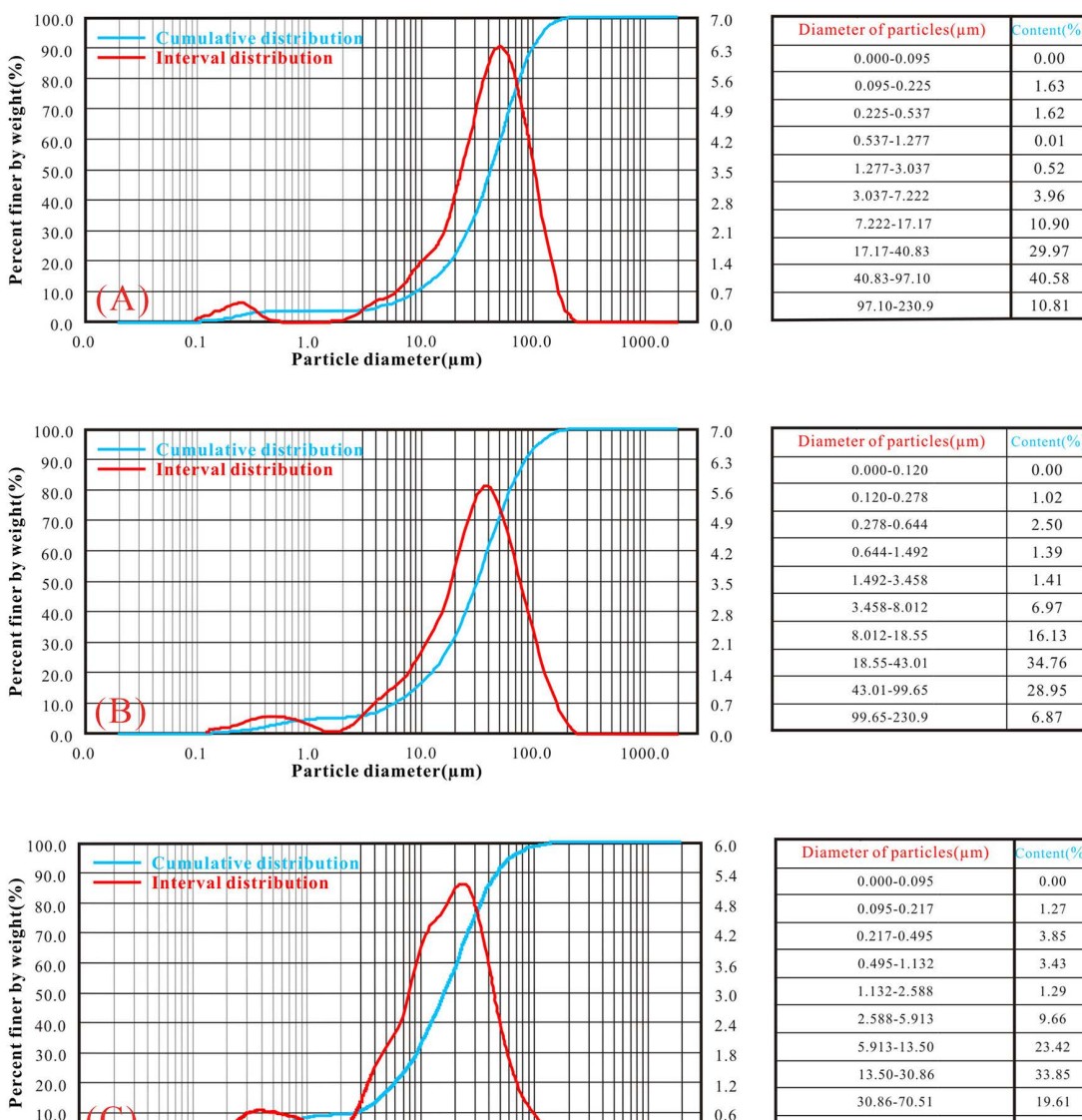

**Fig 4. Particle size distribution of loess.** (A) sandy loess (Jingbian County, northwestern Chinese Loess Plateau); (B) silty loess (Yan'an City, central Chinese Loess Plateau); (C) clayey loess (Jingyang County, southern Chinese Loess Plateau).

common mineral composition confirms the unified aeolian genesis of Malan loess, while variations in mineral content are attributed to post-depositional weathering and environmental evolution—consistent with previous aeolian sedimentary studies [32].

## Collapsibility deformation characteristics

Fig 6 illustrates that the collapsibility coefficient of Malan loess first increases and then decreases with increasing axial pressure, with the highest deformation growth rate observed at 200 kPa (defined as the peak collapse pressure). This

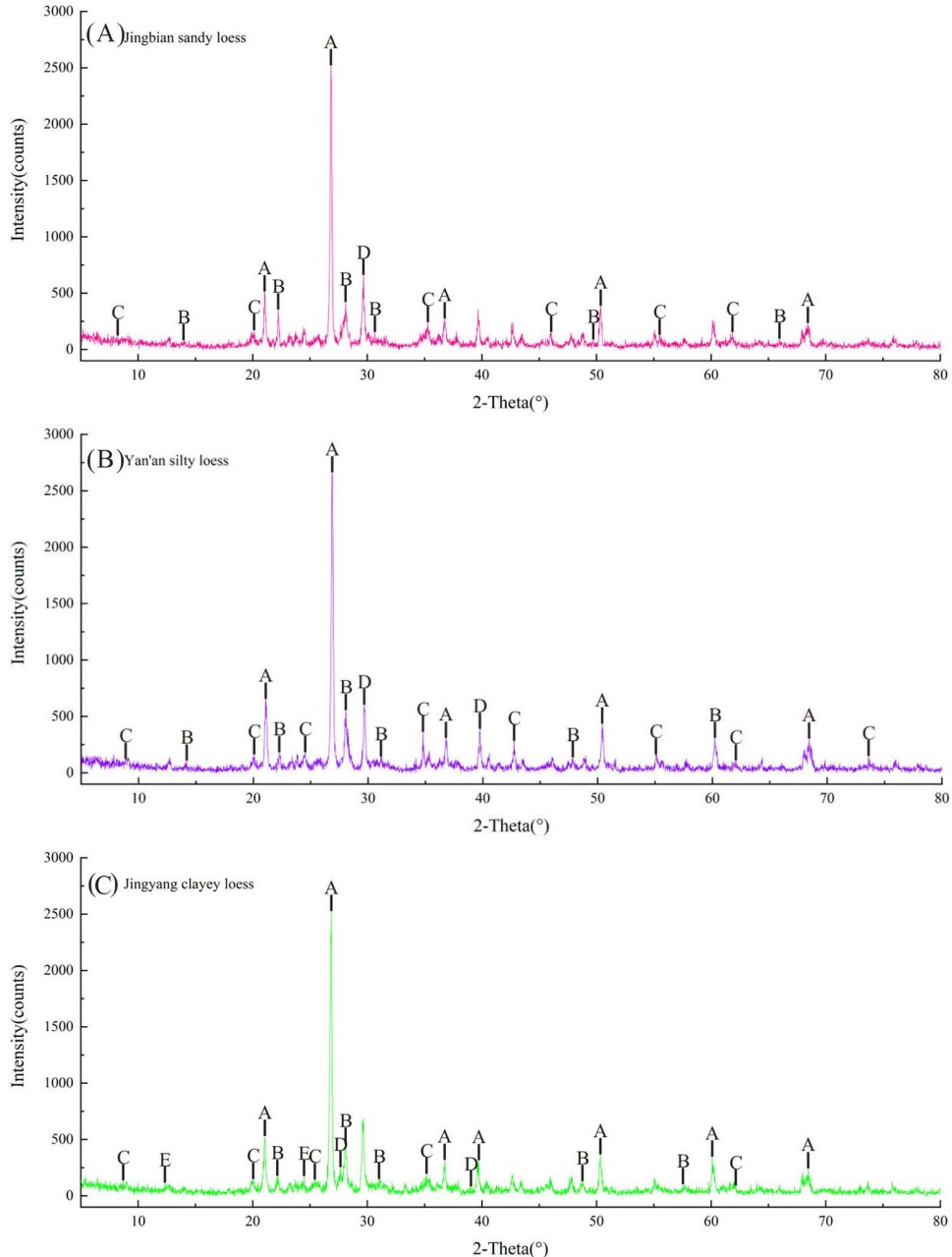

**Fig 5. Test results of loess minerals with different clay content. (A)** Jingbian sandy loess (Jingbian County, northwestern Chinese Loess Plateau) A: Quartz; B: Albite; C: Muscovite; D: Calcite; **(B)** Yan'an silty loess (Jingyang County, southern Chinese Loess Plateau) A: Quartz; B: Albite; C: Muscovite; D: Calcite; **(C)** Jingyang clayey loess (Jingyang County, southern Chinese Loess Plateau) A: Quartz; B: Albite; C: Muscovite; D: Calcite; E: Kaolinite.

trend indicates that the axial collapse deformation also increases first and then decreases with increasing axial pressure, a typical mechanical response of collapsible loess under graded loading. Among the three loess types, collapsibility decreases in the order of sandy loess > silty loess > clayey loess, with peak collapse coefficients of 0.0678, 0.0646, and 0.0519, respectively—all classified as medium collapsibility soil according to relevant geotechnical standards. Differences

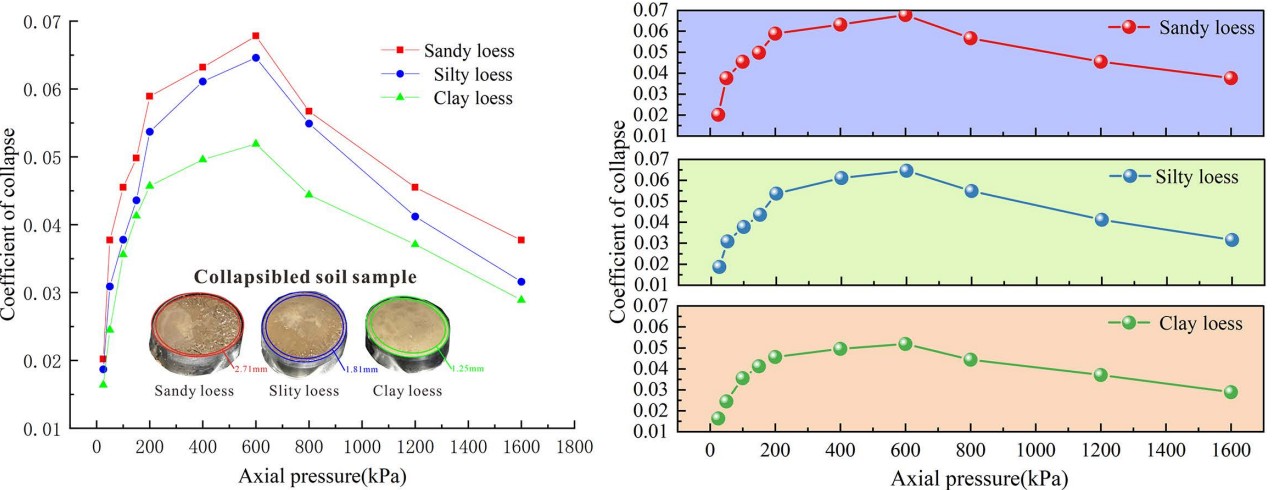

**Fig 6. Collapsibility coefficient of Malan loess with different particle sizes. (A)** Curves of collapsibility coefficient varying with axial pressure and sample schematic diagram; **(B)** Comparison chart of curves showing the relationship between collapsibility coefficient and axial pressure.

in collapsibility coefficients are negligible under low axial pressure (<100 kPa) but become significant under high axial pressure (>200 kPa).

This variation in collapsibility is dominated by the coupling effect of soil skeleton compaction and water-induced cementation degradation under axial pressure, which is inherently determined by the microstructural traits of the loess and can be elaborated in two stages:

### 1. Low-to-medium axial pressure stage (< 600 kPa)

Sandy loess possesses a loose skeleton-supported structure, abundant open pores, and weak interparticle cementation. With increasing axial pressure, the loose skeleton is gradually compacted, and particle contact stress rises; after water immersion, soluble cementitious components soften and dissolve rapidly, triggering pore collapse and particle rearrangement, making it prone to particle rearrangement and pore collapse under pressure and thereby leading to higher collapsibility. Clayey loess is characterized by a high content of fine-grained particles (particle size<20μm, accounting for 59%), which exerts a dual control on its collapsibility behavior [33]. On the one hand, fine particles fill the intergranular pores and form a dense micro-pore network, reducing the effective permeability of the soil; during water immersion, the slow water infiltration rate delays the softening of cementitious components, thus inhibiting the initiation of collapse deformation. On the other hand, fine particles (mainly clay minerals) enhance the interparticle cementation via surface adsorption and chemical bonding, forming a more stable aggregate structure that resists the breakage and sliding of soil particles under pressure. Clayey loess, with its high clay content, strong interparticle cementation, and dense structure, exhibits enhanced deformation resistance, resulting in lower collapsibility. Silty loess, as a transitional type, displays moderate collapsibility due to its balanced particle composition and gradation.

### 2. High axial pressure stage (> 600 kPa)

When axial pressure exceeds 200 kPa—the critical threshold for the destruction of the loess's original structural stability—the original open pores in the loess are fully compacted, and particles interlock tightly to form a dense structure via interparticle friction and mechanical embedding, significantly enhancing the bearing capacity of the soil skeleton. For clayey loess, the high fine particle content further promotes the mechanical interlocking effect under high pressure; the fine

particles fill the residual pores between coarse particles, forming a denser soil matrix with fewer defects. Even after water immersion, only minor micro-pore closure deformation occurs; water can barely penetrate the dense particle arrangement to weaken interparticle bonds, and strong interparticle interlocking effectively inhibits the development of collapse deformation. Consequently, the soil is gradually densified, the axial collapse deformation decreases, the collapsibility coefficient declines with further increased axial pressure, and the growth rate of collapsibility slows down—an observation that provides critical insights for engineering design in loess regions.

## Microstructural variation characteristics of Malan loess before and after collapsibility

**Qualitative Microstructural Observations (SEM).** To improve the transparency of the experimental procedure and ensure the representativeness and objectivity of scanning electron microscopy (SEM) observations, three observation fields were randomly selected from the central region of each sample for all types of loess (sandy loess, silty loess, and clayey loess). Representative fields were chosen to be free of obvious cracks and impurity interference, while truly reflecting particle morphology, clay distribution, particle contacts, and pore characteristics.

Scanning electron microscopy (SEM) micrographs of the three loess types are presented in Fig 7. In terms of particle morphology, sandy loess particles are angular/sub-angular, silty loess particles are lamellar/rod-like, and clayey loess particles are sub-rounded/rounded. Clay particle distribution also differs across the three types: sandy loess clay adheres to coarse particles (forming "clay coatings"), silty loess clay acts as a cementing agent between particles, and clayey loess clay fills interparticle voids. Particle contact modes include point-to-point for sandy loess, point-to-edge for silty loess, and edge-to-edge for clayey loess, while pore types are dominated by open pores in sandy loess, open/occluded pores in silty loess, and cemented pores in clayey loess. After collapsibility (Fig 8), significant microstructural changes occur: Sandy loess open pores are nearly eliminated, with only a few occluded/cemented pores remaining, a phenomenon caused by water-induced softening of clay coatings that triggers particle rearrangement and open-pore collapse. Silty loess open/occluded pores are converted to occluded pores as water destroys the open-pore structure and induces compressive deformation. Clayey loess cementation weakens, and detached fine particles fill pores, resulting from clay softening and dispersion upon water immersion. These changes directly explain the macroscopically observed collapsibility differences: the looser the original structure and the higher the proportion of open pores, the more significant the collapsible deformation.

## Quantitative microstructural analysis (PCAS)

To ensure the reliability and repeatability of PCAS quantitative analysis, the same sample selection criteria as those used for SEM observations were adopted. Three SEM observation fields were selected for each sample, and key parameters including pore area ratio and pore gradation were automatically extracted and statistically analyzed by the system software. The average values of these parameters were used as the final quantitative indicators to minimize observation errors caused by a single field of view.

Key microstructural parameters were analyzed using the PCAS system [34] (Table 2). The pore area ratio of the three loess types is similar before collapse; after collapse, it decreases by 35.36% for sandy loess, 33.11% for silty loess, and 32.12% for clayey loess. All undisturbed samples have the highest mesopore proportion and lowest macropore proportion; after collapse, mesopores increase by ~8%, macropores decrease by >80%, and micropores slightly decrease—consistent with previous studies [32], which validates the reliability of the experimental results. Pore shape factors increase modestly after collapse, indicating reduced pore boundary irregularity, while probability entropy decreases and pore abundance increases, confirming more regular pore shapes. Pore curvature coefficients [35] increase, enhancing pore channel tortuosity and potentially improving water-retention performance. These quantitative changes collectively demonstrate that collapsible deformation restructures the loess's pore system: macropores (prone to collapse) are converted to mesopores, and pore morphology becomes simpler—directly reflecting particle rearrangement and compaction.

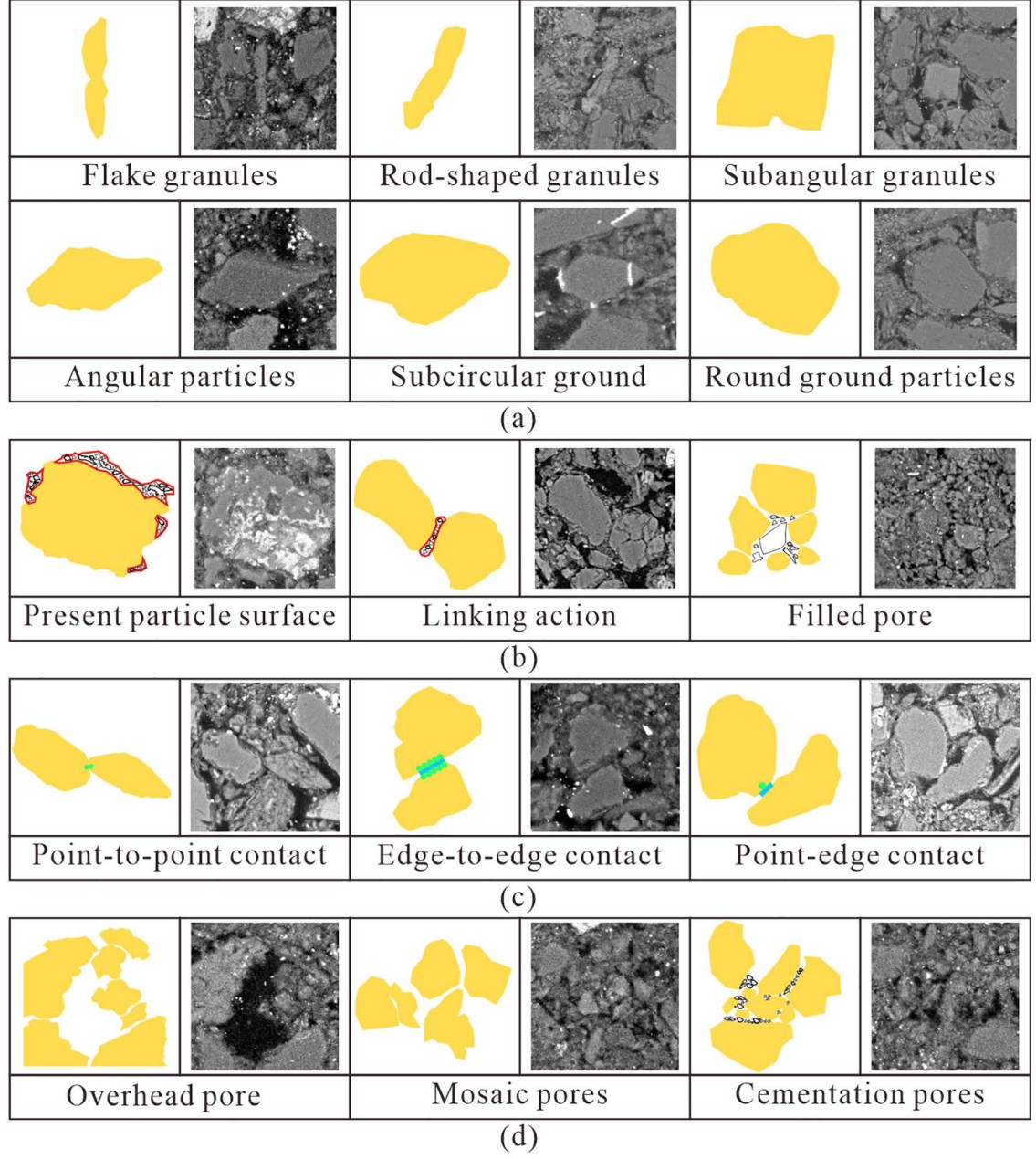

**Fig 7. Microstructure characteristics of Malan loess with different particle sizes. (A)** Particle morphology; **(B)** Clay particle distribution; **(C)** Inter-particle contact; **(D)** Interparticle porosity.

## Directional frequency

Fig 9(A) shows that pore orientation frequency [36] in sandy loess and Yan'an silty loess are concentrated in the 0°–45° and 135°–180° ranges, while clayey loess pores are focused on 45°–135°. After collapse, pores in horizontal (0°/180°) and vertical (90°) directions are preferentially eliminated, with more pores remaining at 45° and 135°. This indicates that horizontal and vertical pores are structurally unstable and more susceptible to deformation under the combined action of

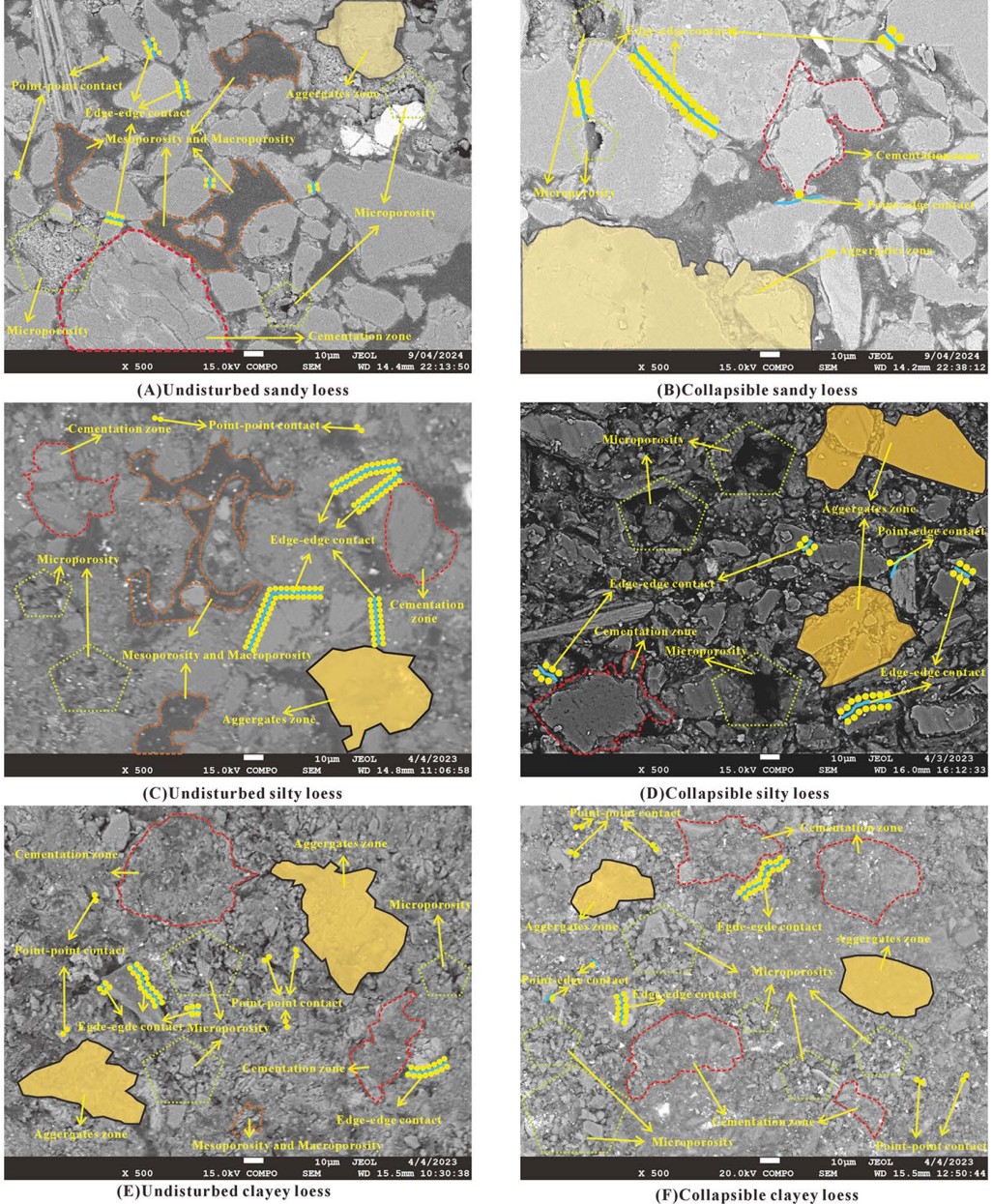

**Fig 8. Microstructural changes of Malan loess with different particle sizes. (A)**Undisturbed sandy loess; **(B)**Collapsible sandy loess; **(C)**Undisturbed silty loess; **(D)**Collapsible silty loess; **(E)**Undisturbed clayey loess; **(F)**Collapsible clayey loess.

pressure and water—providing a microscopic explanation for the vertical settlement and horizontal deformation commonly observed in loess engineering due to collapsibility.

## Fractal dimension

Fig 9(B) and (C) show a strong linear correlation (R²>0.95) between the particle and pore fractal dimension curves of Jingbian sandy loess, confirming the fractal nature of its microstructure. Fig 9(D) and (E) demonstrate that the particle

**Table 2. Pore area ratio of Malan loess with different particle sizes.**

| Sample Type | Pore area ratio | Pore size distribution | | | Form Factor | Probability Entropy | Pore abundance | Curvature Coefficient |
|---|---|---|---|---|---|---|---|---|
| | | Micropore | Mesopore | Macropore | | | | |
| Undisturbed Sandy Loess | 47.8% | 40.2% | 54.6% | 5.2% | 0.328 | 0.991 | 0.525 | 0.968 |
| Undisturbed Silty Loess | 45.6% | 36.9% | 58.3% | 4.8% | 0.318 | 0.986 | 0.541 | 1.046 |
| Undisturbed Clayey Loess | 43.9% | 35.5% | 60.0% | 4.5% | 0.357 | 0.983 | 0.573 | 1.105 |
| Collapse Sandy Loess | 30.9% | 35.5% | 63.4% | 1.1% | 0.375 | 0.901 | 0.635 | 1.419 |
| Collapse Silty Loess | 30.5% | 32.6% | 66.4% | 1.0% | 0.381 | 0.850 | 0.663 | 1.324 |
| Collapse Clayey Loess | 29.8% | 31.3% | 67.8% | 0.9% | 0.495 | 0.844 | 0.680 | 1.153 |

fractal dimensions of Jingbian and Jingyang loess decrease after collapse, an outcome attributed to changed particle contact modes (e.g., from point-to-point to point-to-edge) that simplify particle boundaries. This contact mode transition is inherently regulated by the initial morphological characteristics of loess particles, as supported by relevant studies [37,38]. For instance, angular and subangular particles with irregular edges tend to form initial point-to-point contact due to their sharp protrusions; after collapse, the abrasion of edges and particle fragmentation transform the contact into point-to-edge or edge-to-edge contact. In contrast, rounded and subrounded particles initially exhibit more stable point-to-point or surface-to-surface contact, which transitions to point-to-edge contact as collapse-induced deformation occurs. Platy and rod-like particles, characterized by their anisotropic shapes, are prone to surface-to-edge or edge-to-edge contact even before collapse, with the contact interface becoming more simplified and regular after collapse due to crushing and rearrangement.

The pore fractal dimensions of all three types decrease significantly, reflecting the transformation from complex to simple pore boundaries (macropores to mesopores), which is consistent with the pore size distribution results. This pore structure evolution in loess is closely related to its intrinsic pore characteristics and collapse-induced particle rearrangement: loess inherently contains abundant interparticle macropores and intraparticle micropores, and during collapse, the compression of loose interparticle macropores and the closure of intraparticle micropores (caused by particle crushing and cementation degradation) jointly simplify the pore network. Similar to the dominant role of large pores in calcareous sand [39], the reduction of macropores in loess is the primary contributor to the decrease in pore fractal dimension, as macropores account for the majority of pore volume and their transformation directly alters the complexity of the pore boundary. In contrast, the change in micropores has a negligible effect on the overall pore fractal dimension due to their small volume fraction.

## Microscopic collapsibility mechanisms of malan loess with different particle sizes

The disparities in collapsibility and microstructural evolutions among the three loess categories are inherently attributed to their genetic origins and particle size distributions. Sandy loess, deposited aeolianly under arid climatic regimes, is characterized by coarse angular particles, low clay content, and well-developed open pores. The water-induced softening of clay coatings serves as a lubricating agent for particles, initiating open-pore collapse and particle rearrangement—which constitutes its dominant collapsibility mechanism. Silty loess, also an aeolian deposit formed under arid conditions, exhibits heterogeneous lamellar/rod-like particles, moderate clay content, and predominant point-to-edge contact modes. Upon water immersion, it undergoes open-pore collapse, macropore-to-mesopore transformation, and intensified particle rearrangement, consequently manifesting moderate collapsibility. Clayey loess, generated via long-distance aeolian transport under humid climatic environments, consists of fine sub-rounded particles, high clay content, and prevalent edge-to-edge

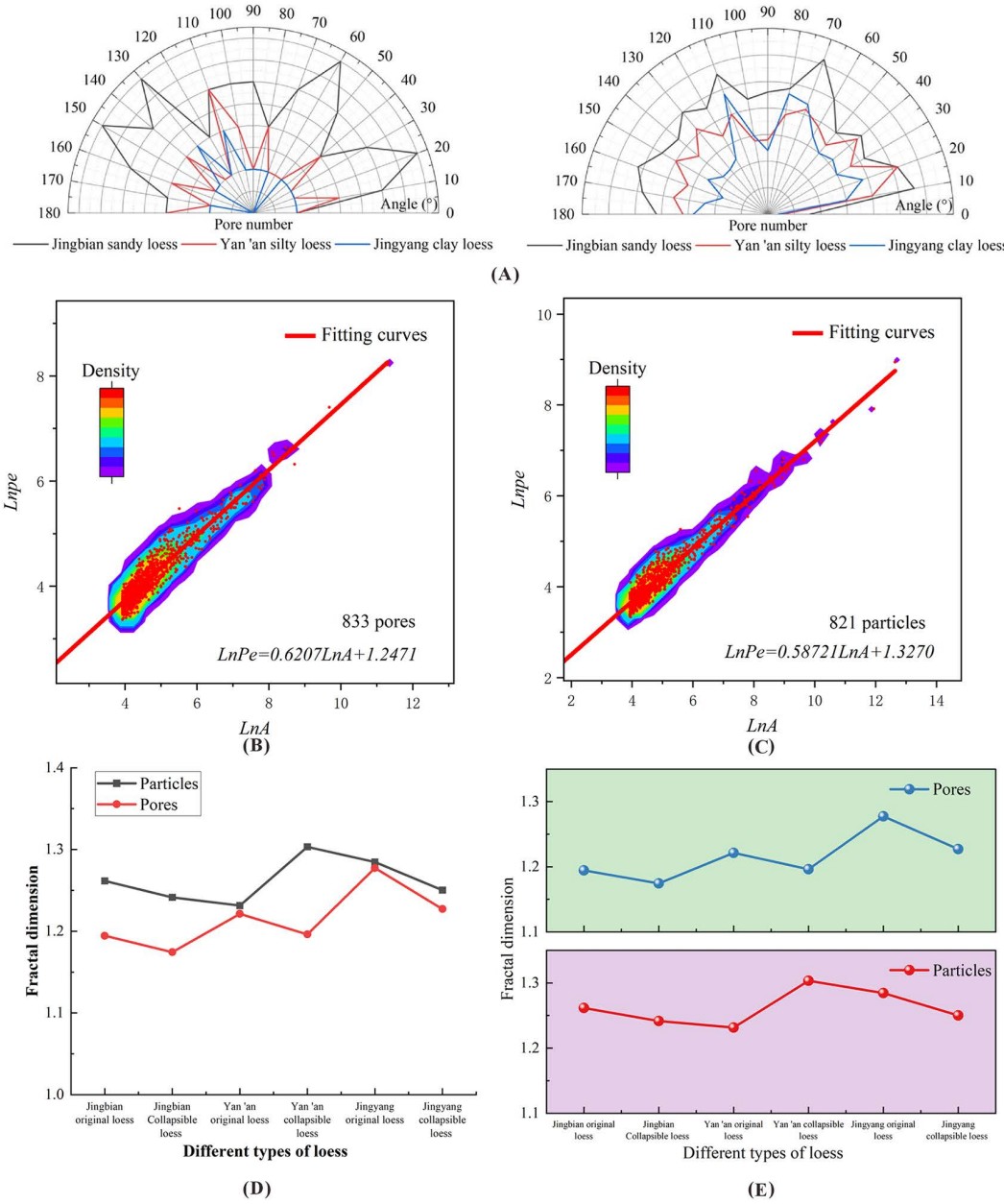

**Fig 9. Pore direction frequency and fractal dimension of Malan loess with different particle sizes. (A)** Pore direction frequency; **(B)** Fractal dimension of pores after collapsibility of sand loess; **(C)** Fractal dimension of particles after collapsibility of sand loess; **(D)** Curves of pore and particle fractal dimensions varying with sample type; **(E)** Comparison chart of curves of pore and particle fractal dimension changes.

contacts. Water saturation triggers pore collapse, clay softening/dispersion, and pore filling, ultimately leading to hydraulic settlement.

Based on the aforementioned findings, a "particle-pore-clay synergistic collapsibility mechanism model" (Fig 10) is proposed. This model not only establishes an intrinsic correlation between loess genesis, particle size characteristics, microstructure, and collapsibility but also reveals the inherent evolutionary law of "genetic endowment → initial structural

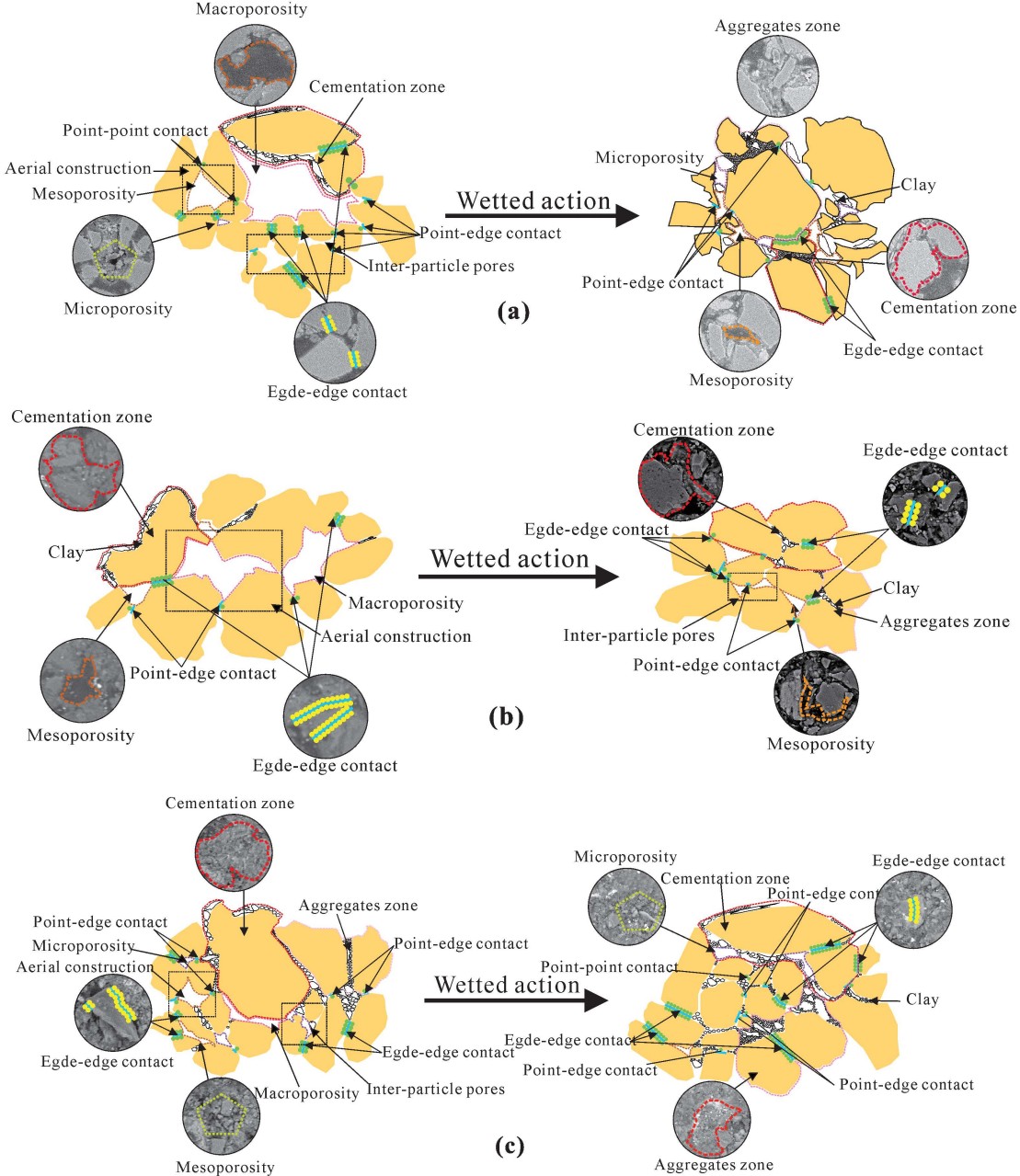

**Fig 10. Generalized collapse model of Malan loess with different particle sizes. (A)** Jingbian sandy loess; **(B)** Yan'an silty loess; **(C)** Jingyang clayey loess.

differentiation → synergistic evolution during collapsibility". Distinct from existing theories that focus on single-factor effects, this model integrates genetic endowment (genesis-driven variations in particle morphology/size), physicochemical behaviors of clay (hydration, softening, and cementation degradation), and pore evolution (macropore reduction – mesopore expansion – micropore reconstruction) into a coupled dynamic system. Its innovative academic contributions and

distinctions from existing theories are elaborated as follows: First, it breaks through the limitation of traditional structural theories that solely emphasize particle skeleton transformation, highlighting the "mutual triggering - dynamic feedback - synergistic driving" interaction among particles, pores, and clay, and clarifies that the physicochemical behavior of clay acts as the core driver of collapsibility rather than a passive component. Second, differing from the fine-particle filling theory, it defines clay as a "chemical-mechanical coupling medium", whose hydration and dispersion behaviors directly regulate the transformation of particle contact modes (from point-point/point-edge to point-surface/surface-surface) and the topological evolution of pores. Third, it advances pore evolution theories by establishing quantitative correlations between microscopic structural parameters (particle contact mode, clay cementation state) and macroscopic collapsibility indicators (collapsibility coefficient, settlement magnitude).

This model provides a unified theoretical framework for interpreting the collapsibility disparaties among diverse loess types and offers scientific support for the understanding, prediction, and mitigation of loess collapsibility in engineering practices, particularly in regions with complex loess genetic backgrounds [40]. For foundation pretreatment, the key focus lies on initial clay content, macropore proportion, and particle contact mode: targeted compaction is employed for sandy loess to inhibit open-pore collapse, while chemical grouting is implemented for clayey loess to enhance inter-particle cementation and prevent clay dispersion. For the prediction of slope collapsible hazards, slope gradient, pore evolution law, and clay cementation state are integrated to predict collapsible deformation under rainfall infiltration, supplemented by supporting measures such as slope surface waterproofing, internal drainage systems, and localized anchor reinforcement to mitigate the risks of rapid collapse and progressive sliding. For the optimization of underground structure stability, pore heterogeneity, clay content, and excavation-induced stress redistribution are taken into account: advance support parameters (e.g., bolt length and spacing) are designed for sandy loess strata to constrain pore collapse and particle rearrangement, while the form of lining structures is optimized for clayey loess strata to adapt to stress release caused by cementation layer destruction, thereby reducing the risk of lining cracking induced by uneven collapsible deformation.

## Conclusions

In this paper, the collapsibility test of three kinds of Malan loess with different particle sizes was conducted, the microstructure and parameter changes of the three kinds of loess before and after collapsibility were analyzed, and the collapsibility mechanism of the three kinds of loess was proposed, and the following conclusions were drawn:

1. The regional differentiation laws of particle composition and physical-mechanical properties of Malan loess were clarified: from the northwest to the southeast, the clay content increases (8.67%–27.87%), and parameters such as natural density and liquid limit show a regular increasing trend, providing a quantitative basis for the prediction of loess engineering characteristics;

2. The nonlinear relationship between axial pressure and collapsible deformation was revealed, and 200 kPa was determined as the critical value of peak collapse pressure. Moreover, the collapsibility follows the order of sandy loess > silty loess > clayey loess, which supplements the pressure threshold parameters for the collapsibility evaluation of loess with different particle sizes;

3. The coupling relationship between particle morphology (angular → subrounded → rounded), clay distribution patterns (adherent → cemented → filled) and collapse mechanism was illustrated, perfecting the micro-driving theory of loess collapsibility;

4. A "particle-pore-clay synergistic collapse model" was proposed, which establishes the internal correlation among genetic endowment, microstructure and macroscopic collapsibility, breaking through the limitations of traditional single-factor theories.

## Supporting information

**S1 Fig. Raw picture of microstructure changes of Malan loess with different particle sizes.** (A) Undisturbed sandy loess; (B) Collapsible sandy loess; (C) Undisturbed silty loess; (D) Collapsible silty loess; (E) Undisturbed clayey loess; (F) Collapsible clayey loess.
(PDF)

## Acknowledgments

Thanks to all those who provided guidance, support and encouragement during the research process. Your help has enabled the completion of this work.

## Author contributions

**Conceptualization:** Li Li, Qinglong Zhang, Yisong Bai.

**Data curation:** Yisong Bai.

**Formal analysis:** Huandong Mu.

**Funding acquisition:** Huandong Mu.

**Investigation:** Li Li.

**Methodology:** Huandong Mu.

**Software:** Longhao Zheng.

**Validation:** Longhao Zheng, Yisong Bai.

**Writing – original draft:** Longhao Zheng.

**Writing – review & editing:** Huandong Mu, Yisong Bai.

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
