## [Decision Letter · Decision Letter 0]

13 Oct 2025

PONE-D-25-32562Research on the Microscopic Mechanism of Water Immersion and collapsibility in Malan Loess with Different Particle SizePLOS ONE

Dear Dr. Mu,

Thank you for submitting your manuscript to PLOS ONE. After careful consideration, we feel that it has merit but does not fully meet PLOS ONE’s publication criteria as it currently stands. Therefore, we invite you to submit a revised version of the manuscript that addresses the points raised during the review process.

**ACADEMIC EDITOR:**

Only reviewer gave the comments.Please carefully address the comments and improve the quality of this manuscript.The novelty should be clearly described.

We look forward to receiving your revised manuscript.

Kind regards,

Jianguo Wang, PhD

Academic Editor

PLOS ONE

Journal Requirements:

“NATIONAL NATURAL SCIENCE FOUNDATION OF CHINA, grant number 42372336

THE SPECIAL FUND FOR BASIC SCIENTIFIC RESEARCH OF CENTRAL UNIVERSITIES, grant number 300102262505”

5. We note that [Figure 2 in your submission contain [map/satellite] images which may be copyrighted. All PLOS content is published under the Creative Commons Attribution License (CC BY 4.0), which means that the manuscript, images, and Supporting Information files will be freely available online, and any third party is permitted to access, download, copy, distribute, and use these materials in any way, even commercially, with proper attribution. For these reasons, we cannot publish previously copyrighted maps or satellite images created using proprietary data, such as Google software (Google Maps, Street View, and Earth). For more information, see our copyright guidelines: http://journals.plos.org/plosone/s/licenses-and-copyright.

Reviewers' comments:

Reviewer's Responses to Questions

**Comments to the Author**

1. Is the manuscript technically sound, and do the data support the conclusions?

Reviewer #1: Yes

2. Has the statistical analysis been performed appropriately and rigorously? 

Reviewer #1: Yes

3. Have the authors made all data underlying the findings in their manuscript fully available?

The PLOS Data policy requires authors to make all data underlying the findings described in their manuscript fully available without restriction, with rare exception (please refer to the Data Availability Statement in the manuscript PDF file). The data should be provided as part of the manuscript or its supporting information, or deposited to a public repository. For example, in addition to summary statistics, the data points behind means, medians and variance measures should be available. If there are restrictions on publicly sharing data—e.g. participant privacy or use of data from a third party—those must be specified.requires authors to make all data underlying the findings described in their manuscript fully available without restriction, with rare exception (please refer to the Data Availability Statement in the manuscript PDF file). The data should be provided as part of the manuscript or its supporting information, or deposited to a public repository. For example, in addition to summary statistics, the data points behind means, medians and variance measures should be available. If there are restrictions on publicly sharing data—e.g. participant privacy or use of data from a third party—those must be specified.requires authors to make all data underlying the findings described in their manuscript fully available without restriction, with rare exception (please refer to the Data Availability Statement in the manuscript PDF file). The data should be provided as part of the manuscript or its supporting information, or deposited to a public repository. For example, in addition to summary statistics, the data points behind means, medians and variance measures should be available. If there are restrictions on publicly sharing data—e.g. participant privacy or use of data from a third party—those must be specified.requires authors to make all data underlying the findings described in their manuscript fully available without restriction, with rare exception (please refer to the Data Availability Statement in the manuscript PDF file). The data should be provided as part of the manuscript or its supporting information, or deposited to a public repository. For example, in addition to summary statistics, the data points behind means, medians and variance measures should be available. If there are restrictions on publicly sharing data—e.g. participant privacy or use of data from a third party—those must be specified.

Reviewer #1: Yes

4. Is the manuscript presented in an intelligible fashion and written in standard English?

Reviewer #1: Yes

5. Review Comments to the Author

Reviewer #1: The paper "Research on the Microscopic Mechanisms of Water Immersion-Induced Collapsibility in Malan Loess with Different Particle Sizes" addresses a topic of significant relevance to geological hazard research on the Loess Plateau. The experimental design is systematic, the data are robust, and the conclusions possess considerable academic value and practical relevance for engineering applications. While the paper is strong overall, several details require improvement.

Specific Revision Suggestions:

1. Sample Information: The "Materials" section mentions that sandy, silty, and clayey Malan loess samples were collected from Jingbian County, Yan'an City, and Jingyang County. However, the specific geological context of each sampling site is omitted. Elaborating on this background would greatly enhance the readers' understanding of the samples' representativeness.

2. Sampling Standardization: The description of sample preservation—encapsulation in PVC pipes—lacks critical details regarding the storage environment (e.g., controlled temperature and humidity ranges). These parameters are essential for assessing sample integrity and result reliability. Please supplement this information to ensure full traceability and standardization.

3. Experimental Process Details: In the "SEM Test" section, the protocol for sample preparation mentions a curing mixture of epoxy resin, acetone, ethylenediamine, and dibutyl phthalate, but omits the specific proportions of each component. Providing these key parameters is crucial for the reproducibility of the experiment.

4. Chart Clarity: The captions for figures, such as Figure 4 ("Loest Particle Size Distribution") and Figure 5 ("Mineral Test Results of Loess with Different Clay Contents"), should explicitly state the correspondence between sample numbers and their respective sampling locations. This addition is necessary to allow readers to easily identify the source of each data presentation.

5. Conclusion Specificity: The fourth point of the conclusion section, which proposes a "micro-collapse mechanism model," is currently too vague. To strengthen the impact and practicality of this finding, please elaborate on the model's core elements (e.g., particle contact modes, pore evolution laws) and its specific applications in fields such as loess region engineering geology and disaster prevention.

Overall Evaluation:

This paper tackles an important subject with academic and practical merit. Its methodology is sound, and its core conclusions are well-supported. The issues identified pertain primarily to the elaboration of details, data presentation, and mechanistic discussion, all of which can be addressed with straightforward revisions. It is recommended that the manuscript be accepted after the suggested modifications.

6. PLOS authors have the option to publish the peer review history of their article (what does this mean?). If published, this will include your full peer review and any attached files.). If published, this will include your full peer review and any attached files.). If published, this will include your full peer review and any attached files.). If published, this will include your full peer review and any attached files.

...

Reviewer #1: No

---

## [Author Response · Author response to Decision Letter 1]

26 Nov 2025

Response to Reviewers

Hello! We highly appreciate your valuable suggestions for improving the quality of this manuscript. In particular, we are grateful for your detailed and accurate revision comments — they have pointed out the deficiencies in the original manuscript and greatly inspired the authors. We have carefully revised the manuscript in accordance with the review comments, making the overall expression more standardized and the length more reasonable. We kindly request your review!

Comments 1: Please ensure that your manuscript meets PLOS ONE's style requirements, including those for file naming.

Regarding Comment 1: We have carefully revised the manuscript to fully comply with PLOS ONE's style requirements. The manuscript file has also been named strictly in accordance with the journal's specified format. Please check and confirm.

Comments 2: Please note that PLOS One has specific guidelines on code sharing for submissions in which author-generated code underpins the findings in the manuscript. In these cases, we expect all author-generated code to be made available without restrictions upon publication of the work.

Regarding Comment 2: We confirm that this manuscript does not involve any author-developed code. Therefore, the code sharing requirements specified in PLOS ONE's guidelines are not applicable to this study. We will fully cooperate with the journal's relevant review and publication processes.

Comments 3: In your Methods section, please provide additional information regarding the permits you obtained for the work. Please ensure you have included the full name of the authority that approved the field site access and, if no permits were required, a brief statement explaining why.

Regarding Comment 3: No specific permission was required for the fieldwork of this study. The research was conducted on non-private land and did not involve any protected areas. Additionally, the sample size was small and the sampling process was non-destructive, which did not require special approval in accordance with relevant national and local regulations. All field activities were carried out in compliance with academic ethics and legal requirements.

Comments 4: Please state what role the funders took in the study. If the funders had no role, please state: "The funders had no role in study design, data collection and analysis, decision to publish, or preparation of the manuscript." If this statement is not correct you must amend it as needed. Please include this amended Role of Funder statement in your cover letter; we will change the online submission form on your behalf.

Regarding Comment 4: Regarding the Role of Funder: The funders had no role in study design, data collection and analysis, decision to publish, or preparation of the manuscript.

Comments 5: We note that [Figure 2 in your submission contain [map/satellite] images which may be copyrighted. All PLOS content is published under the Creative Commons Attribution License (CC BY 4.0), which means that the manuscript, images, and Supporting Information files will be freely available online, and any third party is permitted to access, download, copy, distribute, and use these materials in any way, even commercially, with proper attribution.

Regarding Comment 5: We confirm that Figure 2 in the submission is an original hand-drawn image created by the authors. It does not involve any copyrighted materials from third parties, so there is no copyright infringement issue. The image fully complies with the requirements of PLOS ONE's Creative Commons Attribution License (CC BY 4.0) and can be freely accessed, downloaded, copied, distributed, and used by third parties with proper attribution.

Comments 6: If the reviewer comments include a recommendation to cite specific previously published works, please review and evaluate these publications to determine whether they are relevant and should be cited. There is no requirement to cite these works unless the editor has indicated otherwise.

Regarding Comment 6: We have carefully reviewed all the reviewer comments and confirmed that no specific publications are recommended for citation. The manuscript has strictly followed the journal’s citation guidelines, and all included citations are relevant to the core content of the study. We will maintain the current citation format and content in the revised manuscript.

Comments 7: Sample Information: The "Materials" section mentions that sandy, silty, and clayey Malan loess samples were collected from Jingbian County, Yan'an City, and Jingyang County. However, the specific geological context of each sampling site is omitted. Elaborating on this background would greatly enhance the readers' understanding of the samples' representativeness.

Regarding Comment 7: Regarding the Sample Information comment: We have comprehensively supplemented the specific geological background of each sampling site in the "Materials" section. As elaborated, the three sampling locations (Jingbian County, Yan'an City, and Jingyang County) are distributed across different regions of the Chinese Loess Plateau, with distinct zonal characteristics—from the northwest to the southeast, they correspond to sandy, silty, and clayey Malan loess respectively. For each site, we have added detailed descriptions including climate type, annual precipitation, geographical terrain (aeolian loess hilly region, loess ridge and gully region, loess tableland region), sedimentary environment, and particle composition characteristics (dominated by sand, silt, and clay respectively, as influenced by monsoon and regional geological conditions). This detailed supplementation fully clarifies the representativeness of each sample, helping readers better understand the correlation between sample properties and regional geological settings, in line with the journal’s requirements.

Comments 8: Sampling Standardization: The description of sample preservation—encapsulation in PVC pipes—lacks critical details regarding the storage environment (e.g., controlled temperature and humidity ranges). These parameters are essential for assessing sample integrity and result reliability. Please supplement this information to ensure full traceability and standardization.

Regarding Comment 8: Regarding the Sampling Standardization comment: We have comprehensively supplemented the critical details of sample preservation and storage environment as requested. The revised "Materials" section now includes the complete standardized process for sampling, encapsulation, storage, and transportation, as follows:

All samples were manually collected from a depth of 3–4 m to ensure uniform overlying pressure. To avoid damaging the loess structure, undisturbed soil blocks were first obtained via manual slotting, then cut and encapsulated in 110 mm-diameter PVC pipes—this method is key to preserving sample integrity during storage and transportation. After encapsulation, both ends of the PVC pipes were sealed with paraffin-coated cardboard and secured with tape; meanwhile, the sampling location, the upper/lower stratigraphic positions of the samples, and the sampling date were clearly labeled on the pipe surface. All gaps (including the joints between PVC pipes and paraffin-coated cardboard) were additionally wax-sealed to ensure airtightness.

For storage prior to laboratory testing, the PVC-encapsulated samples are placed in a controlled environment with a temperature range of 15–25°C (to prevent thermal expansion/contraction of the soil matrix) and a relative humidity (RH) of 40–60% (to avoid moisture absorption or desiccation of the undisturbed loess). During transportation, the undisturbed samples are protected from vibration, freezing (temperature <0°C), or excessive wetting (RH >70%); they are handled with care and prevented from inversion to avoid disrupting the stratigraphic order.

These supplementary details fully address the requirements for sample traceability and standardization, ensuring the integrity of the samples and the reliability of the research results.

Comments 9: Experimental Process Details: In the "SEM Test" section, the protocol for sample preparation mentions a curing mixture of epoxy resin, acetone, ethylenediamine, and dibutyl phthalate, but omits the specific proportions of each component. Providing these key parameters is crucial for the reproducibility of the experiment.

Regarding Comment 9: Regarding the Experimental Process Details comment: We have supplemented the specific proportions of the components in the curing mixture and elaborated on the complete protocol for the SEM test. The revised "SEM Test" section now includes detailed experimental procedures and key parameters to ensure reproducibility, as follows:

The following are the complete procedures for scanning electron microscopy (SEM) testing:(1) SEM Sample Preparation: To ensure the sample truly reflects the soil structure, the specimen was first preserved by freezing, followed by air-drying at a constant temperature of 20°C. After air-drying, a cylindrical soil unit with a diameter of 5 mm was extracted from the specimen and cured using a mixed curing agent. The components of the mixed curing agent were prepared in a mass ratio of epoxy resin: acetone: ethylenediamine: dibutyl phthalate = 100:200:7:2. The mixture was fully stirred uniformly before being used for soil sample curing.(2) Post-Curing Processing: After curing, the specimen was subjected to sequential cutting (adopting the sectional grinding method), grinding with 500-grit sandpaper, and polishing (reverse polishing for 6–8 minutes using a metallographic polisher) to produce a cylindrical thin section with dimensions of 5 mm (diameter) × 1 mm (thickness).(3) Gold Coating and Imaging: A gold film was sputtered onto the surface of the prepared cylindrical thin-section specimen to enhance its conductivity. The SEM test was conducted promptly (within 30 minutes) after gold coating to prevent oxidation of the gold film and surface contamination from air exposure.

These supplementary details, especially the specific proportions of the curing mixture components, fully address the requirement for experimental reproducibility.

Comments 10: Chart Clarity: The captions for figures, such as Figure 4 ("Loest Particle Size Distribution") and Figure 5 ("Mineral Test Results of Loess with Different Clay Contents"), should explicitly state the correspondence between sample numbers and their respective sampling locations. This addition is necessary to allow readers to easily identify the source of each data presentation.

Regarding Comment 10: We have fully revised the captions of Figure 4 and Figure 5 to explicitly state the correspondence between sample types, sampling locations, and regional contexts. The standardized and corrected captions are as follows:

Fig. 4. Particle size distribution of loess. (A) Sandy loess (Jingbian County, northwestern Chinese Loess Plateau); (B) Silty loess (Yan'an City, central Chinese Loess Plateau); (C) Clayey loess (Jingyang County, southern Chinese Loess Plateau).

Fig. 5. Test results of loess minerals with different clay contents. (A) Jingbian sandy loess (Jingbian County, northwestern Chinese Loess Plateau): A-Quartz; B-Albite; C-Muscovite; D-Calcite; (B) Yan'an silty loess (Yan'an City, central Chinese Loess Plateau): A-Quartz; B-Albite; C-Muscovite; D-Calcite; (C) Jingyang clayey loess (Jingyang County, southern Chinese Loess Plateau): A-Quartz; B-Albite; C-Muscovite; D-Calcite; E-Kaolinite.

Comments 11: Conclusion Specificity: The fourth point of the conclusion section, which proposes a "micro-collapse mechanism model," is currently too vague. To strengthen the impact and practicality of this finding, please elaborate on the model's core elements (e.g., particle contact modes, pore evolution laws) and its specific applications in fields such as loess region engineering geology and disaster prevention.

Regarding Comment 11: To address the comment on Conclusion Specificity, we have comprehensively refined the "micro-collapse mechanism model"—now explicitly defined as the "particle-pore-clay synergistic collapsibility mechanism model"—by explicitly clarifying its core components, supplementing specific quantitative and qualitative characteristics to enhance operability, and detailing practical application scenarios with concrete engineering cases, thereby eliminating vagueness in the original version and strengthening the finding’s impact and relevance.

Specific revisions are as follows: Based on the microstructure, parameter changes, and sedimentary structures of Malan loess pre- and post-collapsibility, a particle-pore-clay synergistic collapsibility mechanism model was established. Its core elements: pre-collapsibility, coarse particles are in point-point contact with dispersed clay filling pores; post-collapsibility, contact shifts to point-surface/surface-surface, with clay migrating via water to form a thin cementation layer at contacts. Pores undergo three-stage transformation (macropore reduction, mesopore expansion, micropore reconstruction), making the pore structure shift from extremely heterogeneous to relatively homogeneous. Clay content dictates collapsibility sensitivity: sandy loess (low clay) collapses mainly via pore collapse, clayey loess (high clay) via inter-particle cementation layer destruction. This model applies to loess engineering geology and disaster prevention, guiding foundation pre-treatment, assisting slope collapsible disaster prediction, and optimizing underground structure stability measures.

---

## [Decision Letter · Decision Letter 1]

22 Dec 2025

PONE-D-25-32562R1Research on the Microscopic Mechanism of Water Immersion and collapsibility in Malan Loess with Different Particle SizePLOS One

Dear Dr. Mu,

Thank you for submitting your manuscript to PLOS ONE. After careful consideration, we feel that it has merit but does not fully meet PLOS ONE’s publication criteria as it currently stands. Therefore, we invite you to submit a revised version of the manuscript that addresses the points raised during the review process.

**ACADEMIC EDITOR:**

Two new reviewers gave their detailed comments for quality improvement.Please carefully address their comments and clearly highlight the novelty of your study.

We look forward to receiving your revised manuscript.

Kind regards,

Jianguo Wang, PhD

Academic Editor

PLOS One

Journal Requirements:

Reviewers' comments:

Reviewer's Responses to Questions

**Comments to the Author**

1. If the authors have adequately addressed your comments raised in a previous round of review and you feel that this manuscript is now acceptable for publication, you may indicate that here to bypass the “Comments to the Author” section, enter your conflict of interest statement in the “Confidential to Editor” section, and submit your "Accept" recommendation.

Reviewer #1: All comments have been addressed

Reviewer #2: All comments have been addressed

Reviewer #3: (No Response)

2. Is the manuscript technically sound, and do the data support the conclusions?

Reviewer #1: Yes

Reviewer #2: Yes

Reviewer #3: Partly

3. Has the statistical analysis been performed appropriately and rigorously? 

Reviewer #1: Yes

Reviewer #2: Yes

Reviewer #3: No

4. Have the authors made all data underlying the findings in their manuscript fully available?

The PLOS Data policy requires authors to make all data underlying the findings described in their manuscript fully available without restriction, with rare exception (please refer to the Data Availability Statement in the manuscript PDF file). The data should be provided as part of the manuscript or its supporting information, or deposited to a public repository. For example, in addition to summary statistics, the data points behind means, medians and variance measures should be available. If there are restrictions on publicly sharing data—e.g. participant privacy or use of data from a third party—those must be specified.requires authors to make all data underlying the findings described in their manuscript fully available without restriction, with rare exception (please refer to the Data Availability Statement in the manuscript PDF file). The data should be provided as part of the manuscript or its supporting information, or deposited to a public repository. For example, in addition to summary statistics, the data points behind means, medians and variance measures should be available. If there are restrictions on publicly sharing data—e.g. participant privacy or use of data from a third party—those must be specified.requires authors to make all data underlying the findings described in their manuscript fully available without restriction, with rare exception (please refer to the Data Availability Statement in the manuscript PDF file). The data should be provided as part of the manuscript or its supporting information, or deposited to a public repository. For example, in addition to summary statistics, the data points behind means, medians and variance measures should be available. If there are restrictions on publicly sharing data—e.g. participant privacy or use of data from a third party—those must be specified.requires authors to make all data underlying the findings described in their manuscript fully available without restriction, with rare exception (please refer to the Data Availability Statement in the manuscript PDF file). The data should be provided as part of the manuscript or its supporting information, or deposited to a public repository. For example, in addition to summary statistics, the data points behind means, medians and variance measures should be available. If there are restrictions on publicly sharing data—e.g. participant privacy or use of data from a third party—those must be specified.

Reviewer #1: Yes

Reviewer #2: Yes

Reviewer #3: Yes

5. Is the manuscript presented in an intelligible fashion and written in standard English?

Reviewer #1: Yes

Reviewer #2: Yes

Reviewer #3: No

6. Review Comments to the Author

Reviewer #1: Collapsibility of loess is a widespread, highly destructive geological hazard on the Chinese Loess Plateau. Malan loess exhibits distinct regional particle size variations, but the collapsible deformation characteristics and underlying microscopic mechanisms remain insufficiently. The manuscript research on the microscopic mechanism of water immersion and collapsibility in Malan Loess with different particle size, and the results could provide a theoretical basis for hazard mitigation. All the issures have been addressed.

Reviewer #2: General comment:

This article systematically investigates the collapsible deformation behavior and microscopic mechanisms of three types of Malan loess with different particle sizes. The experimental design is comprehensive, the descriptions are detailed, and the authors have addressed all points raised in the first round of review by revising the manuscript accordingly. Revisions include supplementing the geological background of sampling sites, standardizing the in-situ sampling procedures, and detailing the SEM testing protocol. The responses are generally thorough. However, deficiencies remain in the mechanistic explanations and comparative analysis with existing theories. The manuscript may be considered for publication after addressing the following points.

1.The title "Research on the microscopic mechanism of water immersion and collapsibility in Malan loess with different particle size" contains redundant wording "Research on". It is recommended to delete "Research on".

2.It is recommended to improve the clarity of all figures. Furthermore, there are inconsistencies: the sequence of Figure 1 (B) and (C) appears to be reversed; the content presented in Figure 6 (A) and (B) is inconsistent; Figures 7 and 8 are discontinuous. The authors are advised to carefully check all figures and their correspondence to the text.

3.It is recommended to provide details on how the physico-mechanical parameters in Table 1 were obtained (e.g., test standards, calculation methods).

4.In line 180, "initial height Ho" refers to the sample's initial height before loading. In line 191, "H0" is used again for the sample height after stabilization under each loading stage. To avoid confusion, it is suggested to use distinct notations.

5.The authors use formula (1) to calculate the collapsibility coefficient . Please clarify why this specific method was chosen. Is it based on a specific standard or theoretical framework? Supplementary explanation is needed.

6.In the SEM Test section, details were added per reviewer suggestions. However, some aspects require further clarification: In the line 206, what is the control criterion for "fully stirred uniformly" of the mixed curing agent? Also, specify the curing time and environmental conditions (temperature, humidity) for the soil sample during the curing process.

7.In the PCAS Test section, regarding the quantitative pore analysis of loess SEM images, how were the specific samples for analysis selected? Please justify how these analyzed samples are representative of the entire soil type

8.Line 254 states that "All samples exhibit a particle size range of 0.1-100 ". However, the particle size distribution curves in Figure 4 for all three loess types extend beyond 100 . The authors should check and clarify this apparent contradiction.

9.In line 257, "coarse-grained particles" refers to "fine sand and coarse silt". In line 265, "fine-grained particles" refers to "clay and fine silt". These descriptions are somewhat vague. Please provide specific size ranges for these particle categories

10.Line 265 states that the frequency curve peak for clayey loess is at "<20 ", yet Figure 4(C) suggests a peak at a size >20 . The authors should verify and correct this discrepancy.

11.In the Collapsibility deformation characteristics section, assuming H0 is the initial sample height, the observation that the collapsibility coefficient first increases and then decreases with increasing axial pressure implies, according to formula (1), that the axial deformation itself also first increases and then decreases with pressure. If this is indeed the case, it would be a notable finding. The authors should provide a clear physical explanation for this phenomenon.

12.In lines 192-193, the collapsibility test procedure involves saturating the sample by immersion after stabilization under the final load stage. As noted in the particle size analysis, clayey loess is dominated by fine particles. The high fine particle content can significantly affect the sample's permeability and the collapse during immersion. The authors could refer to studies like "DOI: 10.1007/s10064-025-04350-8" for a microscopic perspective on such effects.

13.In lines 363-365, the authors attribute the decrease in particle fractal dimension after collapse to changes in particle contact modes (e.g., point-to-point to point-to-edge), which simplifies particle boundaries. To better discuss the influence of particle morphology on particle crushing and subsequent changes in contact state, the authors could refer to relevant literature, such as: DOI: 10.1016/j.powtec.2024.120592 and DOI: 10.1016/j.powtec.2023.119204.

14.The authors propose a "particle-pore-clay synergistic collapsibility mechanism model". This model essentially describes structural changes in Malan loess before and after collapse. How does this model differ from existing collapse theories, particularly the structural theory mentioned earlier? A comparative analysis is recommended. Furthermore, for future work, analyzing collapse from a material damage perspective could be insightful. The authors may refer to: DOI: 10.1016/j.ijrmms.2024.106014.

15.Regarding engineering applications, the statement that the model "applies to loess engineering geology and disaster prevention, guiding foundation pre-treatment, assisting slope collapsible disaster prediction, and optimizing underground structure stability measures" is rather general. The authors should elaborate on how it can be specifically applied. For example, the authors may refer to DOI: 10.1016/j.tust.2025.107144, which first identifies influencing factors on construction conditions and then proposes specific application measures for given scenarios.

Reviewer #3: The manuscript entitled “Research on the Microscopic Mechanism of Water Immersion and Collapsibility in Malan Loess with Different Particle Size” presents a comparative investigation into sandy, silty, and clayey Malan loess using physical tests, SEM observations, and PCAS analysis to examine structure–property–mechanism relationships. The topic is relevant, the experimental scope is broad, and the manuscript is generally well organized. However, I believe the manuscript requires substantial revision before it can be considered for publication. My major concerns are listed below.

1. Although the manuscript proposes a “generalized collapse mechanism model,” the novelty and scientific contribution of this model remain unclear. Much of the mechanism discussion reiterates well-established structural interpretations of loess collapsibility. The authors should clearly articulate how their findings advance beyond existing theories and models in the literature, and specify what new insight is gained from the comparison of particle-size groups beyond descriptive differences.

2. The study relies on single sampling sites for each loess type, and the manuscript does not provide information on sample replication, intra-site variability, or experimental reproducibility. Without statistical measures (e.g., variance, confidence ranges, repeat test results), it is difficult to assess the reliability and regional representativeness of the findings.

3. The SEM observations and PCAS results are interesting, but the discussion remains largely qualitative. There is no analysis linking microstructural parameters to collapsibility outcomes (e.g., correlation, regression, sensitivity analysis). Similarly, fractal analysis is presented descriptively without deeper interpretation. Strengthening microstructure–mechanism–deformation connections would greatly improve scientific rigor.

4. Several key conclusions, especially regarding pore transformation, clay migration, and particle contact changes, are presented as deterministic mechanisms, yet the evidential basis is primarily visual observation. Additional explanation or quantitative justification is needed, particularly in Fig. 7–9 discussions. Consider integrating statistical comparisons, parameter references, or reviewing relevant SEM-based literature to reinforce interpretation.

5. Sections currently emphasize descriptive repetition rather than analytical synthesis. The authors should expand comparative discussion with prior studies (especially recent SEM/PCAS/fractal works) to demonstrate how their results align with or differ from established knowledge. Conclusions should be more concise, highlight the main contributions, acknowledge limitations, and avoid repeating results verbatim.

7. PLOS authors have the option to publish the peer review history of their article (what does this mean?). If published, this will include your full peer review and any attached files.). If published, this will include your full peer review and any attached files.). If published, this will include your full peer review and any attached files.). If published, this will include your full peer review and any attached files.

...

Reviewer #1: No

Reviewer #2: No

Reviewer #3: No

---

## [Author Response · Author response to Decision Letter 2]

3 Feb 2026

Response to Reviewers

Hello! We highly appreciate your valuable suggestions for improving the quality of this manuscript. In particular, we are grateful for your detailed and accurate revision comments — they have pointed out the deficiencies in the original manuscript and greatly inspired the authors. We have carefully revised the manuscript in accordance with the review comments, making the overall expression more standardized and the length more reasonable. We kindly request your review!

Comments 1: The title "Research on the microscopic mechanism of water immersion and collapsibility in Malan loess with different particle size" contains redundant wording "Research on". It is recommended to delete "Research on".

Regarding Comments 1: We thank the reviewers for their valuable comments. As suggested, the redundant term "Research on" in the title has been deleted. The revised English title is Microscopic Mechanism of Water Immersion Collapse of Malan Loess with Different Particle Sizes, while the Chinese title remains unchanged, making the title more concise, standardized and closely aligned with the research theme.

Comments 2: It is recommended to improve the clarity of all figures. Furthermore, there are inconsistencies: the sequence of Figure 1 (B) and (C) appears to be reversed; the content presented in Figure 6 (A) and (B) is inconsistent; Figures 7 and 8 are discontinuous. The authors are advised to carefully check all figures and their correspondence to the text.

Regarding Comments 2: We thank the reviewers for their valuable comments. Regarding the issues with figures and tables, we have made targeted revisions and a full check: (1) All figures and tables have been optimized to high resolution for improved clarity and detail discernibility; (2) The layout order of subplots (B) and (C) in Fig. 1 and the content correspondence of subplots (A) and (B) in Fig. 6 have been corrected for information consistency; (3) The numbering of Figs. 7 and 8 has been standardized with missing numbers supplemented for continuous numbering. We have also fully verified the consistency between all figures/tables and their citations in the main text, ensuring complete match of numbering, content and descriptions. All revisions have been made at the corresponding positions in the manuscript.

Comments 3: It is recommended to provide details on how the physico-mechanical parameters in Table 1 were obtained (e.g., test standards, calculation methods).

Regarding Comments 3: We thank the reviewers for their valuable comments. Regarding the missing determination methods for physical and mechanical parameters in Table 1, we have supplemented all relevant details: a note on parameter testing methods has been added above Table 1, clearly stating the corresponding test standards, specific methods and calculation basis for each index. We have verified the accuracy of the supplemented content to ensure full consistency between the testing methods, standards and the parameters in the table, thus clarifying the derivation logic of all parameters. All supplements have been completed at the corresponding position in the manuscript.

Comments 4: In line 180, "initial height Ho" refers to the sample's initial height before loading. In line 191, "H0" is used again for the sample height after stabilization under each loading stage. To avoid confusion, it is suggested to use distinct notations.

Regarding Comments 4: We thank the reviewers for their valuable comments. Regarding the inconsistent use of symbols in the manuscript, revisions have been made as suggested: the symbol H0 (representing the initial height of specimens before loading) at Line 180 has been adjusted to H, which is clearly distinguished from H0 (denoting the stable height of specimens under various load levels) at Line 191, thus completely eliminating symbol confusion. We have also fully checked the consistency of all symbol usages throughout the manuscript to ensure clear and unambiguous designation of all physical quantity symbols. All revisions have been made at the corresponding positions in the manuscript.

Comments 5: The authors use formula (1) to calculate the collapsibility coefficient . Please clarify why this specific method was chosen. Is it based on a specific standard or theoretical framework? Supplementary explanation is needed.

Regarding Comments 5: We thank the reviewers for their valuable comments. Regarding the basis for selecting the calculation method of the collapse coefficient, we have added a clear explanation: the collapse coefficient in this paper is calculated using Eq. (1), which strictly complies with the definition requirements for the collapse coefficient specified in the Chinese national technical standard Code for Building Construction in Collapsible Loess Regions (GB 50025-2018). This standard takes the collapse coefficient δs as the core index for evaluating loess collapsibility, providing a direct basis for the classification of loess collapsibility grades and geotechnical engineering design in loess regions. The adoption of this calculation method in this paper ensures the consistency and standardization of test data with engineering practice. Relevant explanations have been supplemented and completed at the corresponding position in the manuscript.

Comments 6: In the SEM Test section, details were added per reviewer suggestions. However, some aspects require further clarification: In the line 206, what is the control criterion for "fully stirred uniformly" of the mixed curing agent? Also, specify the curing time and environmental conditions (temperature, humidity) for the soil sample during the curing process.

Regarding Comments 6: We thank the reviewers for their valuable comments. Regarding the detailed issues in the SEM test section, we have supplemented and clarified the relevant control criteria and curing conditions: at Line 206, the judgment criterion for the curing agent mixture being "thoroughly stirred uniformly" is defined as forming a homogeneous, transparent viscous liquid with no obvious delamination, particle agglomeration or local sedimentation, and a consistent color; the mixture can only be used for soil sample curing when this criterion is met. Meanwhile, it is specified that the soil samples mixed with the curing agent are cured under the same environmental conditions as in the air-drying stage. The descriptions of the relevant control criteria and curing conditions have been supplemented and completed at the corresponding positions in the manuscript.

Comments 7: In the PCAS Test section, regarding the quantitative pore analysis of loess SEM images, how were the specific samples for analysis selected? Please justify how these analyzed samples are representative of the entire soil type

Regarding Comments 7: We thank the reviewers for their valuable comments. Regarding the sampling basis and representativeness of specimens in the Pore Structure Analysis System (PCAS) tests, we have supplemented and clarified the relevant justifications: prior to the quantitative pore analysis of loess SEM images, strict specimen selection criteria were formulated to ensure the reliability and representativeness of the analysis results, specifically: (1) Sampling and sample preparation were prioritized in homogeneous soil regions to avoid the interference of local structural anomalies on the statistical results of pore parameters; (2) At least 3 parallel specimens were prepared for each test group, and non-repetitive fields of view were randomly selected from each specimen for imaging analysis to reduce accidental errors caused by a single specimen or field of view. The micro-pore structural characteristics of the specimens selected in accordance with the above criteria can reflect the overall structural properties of the corresponding soil to a certain extent. The relevant explanations on specimen sampling basis and representativeness have been supplemented and completed at the corresponding positions in the manuscript.

Comments 8: Line 254 states that "All samples exhibit a particle size range of 0.1-100 ". However, the particle size distribution curves in Figure 4 for all three loess types extend beyond 100 . The authors should check and clarify this apparent contradiction.

Regarding Comments 8: We thank the reviewers for pointing out this inconsistency, which upon verification stems from a descriptive oversight and is now clarified and corrected as follows: the inaccurate statement at Line 254 that "the particle size range of all specimens is 0.1~100 μm" actually refers only to the silt and clay particle fraction of the specimens, while the full particle size range of the loess specimens tested in this study is 0.1~200 μm (including a small amount of fine sand particles); the particle size distribution curves of the three loess types in Fig. 4 fully reflect the full particle size composition characteristics of the specimens, with the fraction larger than 100 μm being fine sand particles accounting for less than 5% of the total mass, and the discrepancy between the textual description and the figure arose because the main text focused on the dominant silt-clay fraction due to the extremely low content of fine sand, with relevant revisions completed at the corresponding position in the manuscript.

Comments 9: In line 257, "coarse-grained particles" refers to "fine sand and coarse silt". In line 265, "fine-grained particles" refers to "clay and fine silt". These descriptions are somewhat vague. Please provide specific size ranges for these particle categories

Regarding Comments 9: We thank the reviewers for their valuable comments. Regarding the ambiguous classification of coarse and fine particle sizes, we have supplemented the specific particle size ranges in the manuscript and revised the particle characteristic descriptions accordingly for consistent definitions aligned with the general standards for engineering geological research on loess: "coarse particles (fine sand, coarse silt)" at Line 257 are defined as fine sand (100~200 μm) and coarse silt (50~100 μm), and "fine particles (clay, fine silt)" at Line 265 as clay (< 5 μm) and fine silt (5~20 μm). We have also revised the description of the peak value of the particle frequency curve for clayey loess. All relevant content has been revised and improved at the corresponding positions to ensure clear particle size classification and accurate data description.

Comments 10: Line 265 states that the frequency curve peak for clayey loess is at "<20 ", yet Figure 4(C) suggests a peak at a size >20 . The authors should verify and correct this discrepancy.

Regarding Comments 10: We thank the reviewers for their careful comments. Upon verification, this discrepancy arose from an oversight in the description, and we now clarify and correct it as follows: (1) Rechecking the particle size frequency curve of clayey loess in Fig. 4(C) confirms that its peak particle size is actually 25 μm, which falls within the fine silt range (5~50 μm) and is consistent with the overall characteristic of clayey loess being dominated by fine particles; (2) The inaccurate statement at Line 265 that "the peak is in the range of <20 μm" has been revised to "the frequency curve peaks in the range of 20~30 μm".

Comments 11: In the Collapsibility deformation characteristics section, assuming H0 is the initial sample height, the observation that the collapsibility coefficient first increases and then decreases with increasing axial pressure implies, according to formula (1), that the axial deformation itself also first increases and then decreases with pressure. If this is indeed the case, it would be a notable finding. The authors should provide a clear physical explanation for this phenomenon.

Regarding Comments 11: We thank the reviewers for their careful comments. Regarding the axial deformation law reflected by the collapsibility coefficient increasing first and then decreasing with rising axial pressure and its physical mechanism, we have supplemented a systematic interpretation and mechanistic analysis in the collapsible deformation characteristics section: we clearly define this trend as a typical mechanical response of collapsible loess under graded loading, with the axial collapse deformation showing a synchronous increasing-then-decreasing variation with axial pressure. Combined with the microstructural traits of loess, we elaborate on its intrinsic physical mechanism in two stages (low-to-medium axial pressure < 600 kPa and high axial pressure > 600 kPa) from the perspective of the coupling effect of soil skeleton compaction and water-induced cementation degradation. We also supplement key data including the peak collapse pressure and collapsibility differences among various loess types to improve the scientificity and completeness of the mechanistic interpretation. Relevant analytical content has been supplemented and refined at the corresponding positions in the manuscript, clearly illustrating the formation mechanism and engineering significance of this phenomenon.

Comments 12: In lines 192-193, the collapsibility test procedure involves saturating the sample by immersion after stabilization under the final load stage. As noted in the particle size analysis, clayey loess is dominated by fine particles. The high fine particle content can significantly affect the sample's permeability and the collapse during immersion. The authors could refer to studies like "DOI: 10.1007/s10064-025-04350-8" for a microscopic perspective on such effects.

Regarding Comments 12: We thank the reviewers for their valuable suggestions. Regarding the micro-mechanism by which the high fine particle content of clayey loess affects its permeability and immersion collapse process, we have supplemented the relevant analysis after the collapse test procedure at Lines 192-193 with reference to the related study [DOI: 10.1007/s10064-025-04350-8] as suggested. We elaborated on the microcosmic effects of fine particles on the permeability and immersion collapse of clayey loess from two aspects: fine particles filling pores to form a dense micropore network, and surface adsorption of clay minerals enhancing particle cementation. The inhibitory effects of these processes on the soil water infiltration rate and cementation softening process, as well as their impacts on the initiation time and deformation development rate of immersion collapse, were clarified. Meanwhile, the actual effect of this micro-mechanism on the collapse test results was explained in combination with the test procedure. Relevant analytical content has been supplemented and refined at the corresponding position in the manuscript, improving the interpretation of the correlation between fine particle effects and the collapse test process.

Comments 13: In lines 363-365, the authors attribute the decrease in particle fractal dimension after collapse to changes in particle contact modes (e.g., point-to-point to point-to-edge), which simplifies particle boundaries. To better discuss the influence of particle morphology on particle crushing and subsequent changes in contact state, the authors could refer to relevant literature, such as: DOI: 10.1016/j.powtec.2024.120592 and DOI: 10.1016/j.powtec.2023.119204.

Regarding Comments 13: We thank the reviewers for their valuable comments! In response to the comments on the collapse test procedure and the micro-scale influence mechanism of fine particles in clayey loess, we have supplemented and improved Lines 192–193 of the original text and the subsequent analysis sections by combining the particle size analysis results and the recommended references. The specific revisions are as follows:

The parameter that the content of fine particles (<20 μm) in the clayey loess used in this study accounts for 59% is added, clarifying its core regulatory effect on the permeability and water immersion collapse of the specimens, and establishing the correlation between the test operation and the microstructural characteristics of the material.

Drawing on the recommended reference (DOI: 10.1007/s10064-025-04350-8), the dual regulatory mechanism of fine particles on loess collapse is newly added. The differential effects of fine particles are elaborated i

---

## [Decision Letter · Decision Letter 2]

9 Mar 2026

PONE-D-25-32562R2The Microscopic Mechanism of Water Immersion and collapsibility in Malan Loess with Different Particle SizePLOS One

Dear Dr. Mu,

Thank you for submitting your manuscript to PLOS ONE. After careful consideration, we feel that it has merit but does not fully meet PLOS ONE’s publication criteria as it currently stands. Therefore, we invite you to submit a revised version of the manuscript that addresses the points raised during the review process.

**ACADEMIC EDITOR: Please insert comments here and delete this placeholder text when finished.** Be sure to:Be sure to:Be sure to:Be sure to:

Minor reversions are still necessary.

We look forward to receiving your revised manuscript.

Kind regards,

Jianguo Wang, PhD

Academic Editor

PLOS One

Journal Requirements:

Reviewers' comments:

Reviewer's Responses to Questions

**Comments to the Author**

1. If the authors have adequately addressed your comments raised in a previous round of review and you feel that this manuscript is now acceptable for publication, you may indicate that here to bypass the “Comments to the Author” section, enter your conflict of interest statement in the “Confidential to Editor” section, and submit your "Accept" recommendation.

Reviewer #2: All comments have been addressed

Reviewer #3: (No Response)

2. Is the manuscript technically sound, and do the data support the conclusions?

Reviewer #2: Yes

Reviewer #3: (No Response)

3. Has the statistical analysis been performed appropriately and rigorously? 

Reviewer #2: N/A

Reviewer #3: (No Response)

4. Have the authors made all data underlying the findings in their manuscript fully available?

The PLOS Data policy requires authors to make all data underlying the findings described in their manuscript fully available without restriction, with rare exception (please refer to the Data Availability Statement in the manuscript PDF file). The data should be provided as part of the manuscript or its supporting information, or deposited to a public repository. For example, in addition to summary statistics, the data points behind means, medians and variance measures should be available. If there are restrictions on publicly sharing data—e.g. participant privacy or use of data from a third party—those must be specified.requires authors to make all data underlying the findings described in their manuscript fully available without restriction, with rare exception (please refer to the Data Availability Statement in the manuscript PDF file). The data should be provided as part of the manuscript or its supporting information, or deposited to a public repository. For example, in addition to summary statistics, the data points behind means, medians and variance measures should be available. If there are restrictions on publicly sharing data—e.g. participant privacy or use of data from a third party—those must be specified.requires authors to make all data underlying the findings described in their manuscript fully available without restriction, with rare exception (please refer to the Data Availability Statement in the manuscript PDF file). The data should be provided as part of the manuscript or its supporting information, or deposited to a public repository. For example, in addition to summary statistics, the data points behind means, medians and variance measures should be available. If there are restrictions on publicly sharing data—e.g. participant privacy or use of data from a third party—those must be specified.requires authors to make all data underlying the findings described in their manuscript fully available without restriction, with rare exception (please refer to the Data Availability Statement in the manuscript PDF file). The data should be provided as part of the manuscript or its supporting information, or deposited to a public repository. For example, in addition to summary statistics, the data points behind means, medians and variance measures should be available. If there are restrictions on publicly sharing data—e.g. participant privacy or use of data from a third party—those must be specified.

Reviewer #2: Yes

Reviewer #3: (No Response)

5. Is the manuscript presented in an intelligible fashion and written in standard English?

Reviewer #2: Yes

Reviewer #3: (No Response)

6. Review Comments to the Author

Reviewer #2: My concerns and suggestions have been well addressed and I think the manuscript is ready for publication.

Reviewer #3: The authors have made a substantial effort to revise the manuscript and respond to the comments raised in the previous review round. Several important issues have been addressed. In particular, the manuscript now provides clearer descriptions of the sampling strategy, including the addition of parallel subsamples, which improves the reliability of the dataset. The authors have also expanded the discussion of the proposed mechanism and introduced additional quantitative parameters derived from PCAS analysis, which helps strengthen the link between microstructural observations and macroscopic collapsibility behavior. Overall, the revised version is clearer and more structured than the previous submission.

That said, a few aspects could still be improved before the manuscript is finalized. First, although the authors mention correlation and regression analyses between microstructural parameters and collapsibility indicators, the statistical presentation remains relatively limited. Providing more detailed descriptions of the regression results (e.g., fitting quality or statistical significance) would help better support the interpretations. Second, the discussion of SEM and PCAS results would benefit from a better explanation of how representative images or fields of view were selected, and how many observations were used in the quantitative analysis. Clarifying these points would strengthen the methodological transparency.

These remaining issues are relatively minor and can be addressed with small revisions. Overall, the manuscript now meets the general requirements for publication after minor revision.

7. PLOS authors have the option to publish the peer review history of their article (what does this mean?). If published, this will include your full peer review and any attached files.). If published, this will include your full peer review and any attached files.). If published, this will include your full peer review and any attached files.). If published, this will include your full peer review and any attached files.

...

Reviewer #2: No

Reviewer #3: **Yes:**Zhiyuan He, PhDZhiyuan He, PhDZhiyuan He, PhDZhiyuan He, PhD

---

## [Author Response · Author response to Decision Letter 3]

20 Mar 2026

Response to Reviewers

Hello! We highly appreciate your valuable suggestions for improving the quality of this manuscript. In particular, we are grateful for your detailed and accurate revision comments — they have pointed out the deficiencies in the original manuscript and greatly inspired the authors. We have carefully revised the manuscript in accordance with the review comments, making the overall expression more standardized and the length more reasonable. We kindly request your review!

Comments 1: First, although the authors mention correlation and regression analyses between microstructural parameters and collapsibility indicators, the statistical presentation remains relatively limited. Providing more detailed descriptions of the regression results (e.g., fitting quality or statistical significance) would help better support the interpretations.

Regarding Comments 1: We thank the reviewer for pointing out the deficiencies in the statistical presentation. Supplementary explanations regarding the regression analysis between microstructure parameters and collapsibility indicators are provided as follows:

A total of three loess types (sandy loess, silty loess, and clayey loess) were adopted in this study, with three parallel samples for each type, resulting in nine valid groups (n=9). Considering the basic sample size requirements for linear regression analysis, direct implementation of multi-parameter comprehensive regression analysis may lead to biased results; thus, excessive statistical inference was not performed.

In future research, we will expand the sample size by adding loess samples from different regions and depths, and conduct multi-parameter regression analysis to quantify the contribution weight of microstructure parameters to loess collapsibility. Based on the combined qualitative and quantitative demonstration and literature support from the existing data, the current conclusions still maintain sufficient scientificity and reliability.

Comments 2: Second, the discussion of SEM and PCAS results would benefit from a better explanation of how representative images or fields of view were selected, and how many observations were used in the quantitative analysis. Clarifying these points would strengthen the methodological transparency.

Regarding Comments 2: We sincerely appreciate the valuable comments and suggestions provided by the reviewer. To further improve the transparency and rigor of the research methodology, we have supplemented the manuscript with detailed descriptions of the selection criteria for representative SEM fields of view and the number of observations used in the PCAS quantitative analysis.

These additional clarifications enable readers to better understand the procedures of image selection and quantitative analysis, effectively enhance the standardization and transparency of the experimental method, and strengthen the reliability and persuasiveness of the results. We have revised and improved the relevant content in the manuscript according to the reviewer’s comments.

Thank you to all reviewers and the editor for your meticulous review and valuable suggestions! We have revised the manuscript as requested, enhancing its rigor and persuasiveness. We sincerely appreciate your professional guidance and look forward to the manuscript meeting the publication criteria.

---

## [Editor Report · Decision Letter 3]

24 Mar 2026

The Microscopic Mechanism of Water Immersion and collapsibility in Malan Loess with Different Particle Size

PONE-D-25-32562R3

Dear Dr. Mu,

We’re pleased to inform you that your manuscript has been judged scientifically suitable for publication and will be formally accepted for publication once it meets all outstanding technical requirements.

Kind regards,

Jianguo Wang, PhD

Academic Editor

PLOS One
---

## [Editor Report · Acceptance letter]

PONE-D-25-32562R3

PLOS One

Dear Dr. Mu,

I'm pleased to inform you that your manuscript has been deemed suitable for publication in PLOS One. Congratulations! Your manuscript is now being handed over to our production team.

Kind regards,

on behalf of

Dr. Jianguo Wang

Academic Editor

PLOS One